# Emerging Disentanglement in Auto-Encoder Based Unsupervised Image Content Transfer

**Ori Press, Tomer Galanti & Sagie Benaim**
The School of Computer Science
Tel Aviv University
`oripress@mail.tau.ac.il,tomerga2@post.tau.ac.il,sagieb@mail.tau.ac.il`

**Lior Wolf**
Facebook AI Research &
The School of Computer Science
Tel Aviv University
`wolf@fb.com`
`wolf@cs.tau.ac.il`

## Abstract

We study the problem of learning to map, in an unsupervised way, between domains $A$ and $B$, such that the samples $\boldsymbol{b} \in B$ contain all the information that exists in samples $\boldsymbol{a} \in A$ and some additional information. For example, ignoring occlusions, $B$ can be people with glasses, $A$ people without, and the glasses, would be the added information. When mapping a sample $\boldsymbol{a}$ from the first domain to the other domain, the missing information is replicated from an independent reference sample $\boldsymbol{b} \in B$. Thus, in the above example, we can create, for every person without glasses a version with the glasses observed in any face image.

Our solution employs a single two-pathway encoder and a single decoder for both domains. The common part of the two domains and the separate part are encoded as two vectors, and the separate part is fixed at zero for domain $A$. The loss terms are minimal and involve reconstruction losses for the two domains and a domain confusion term. Our analysis shows that under mild assumptions, this architecture, which is much simpler than the literature guided-translation methods, is enough to ensure disentanglement between the two domains. We present convincing results in a few visual domains, such as no-glasses to glasses, adding facial hair based on a reference image, etc.

## 1 Introduction

In the problem of unsupervised domain translation, the algorithm receives two sets of samples, one from each domain, and learns a function that maps between a sample in one domain to the analogous sample in the other domain (Zhu et al., 2017a; Kim et al., 2017; Yi et al., 2017; Benaim & Wolf, 2017; Liu & Tuzel, 2016; Liu et al., 2017; Choi et al., 2017; Conneau et al., 2017; Zhang et al., 2017a;b; Lample et al., 2018). The term unsupervised means, in this context, that the two sets are unpaired.

In this paper, we consider the problem of domain $B$, which contains a type of content that is not present at $A$. As a running example, we consider the problem of mapping between a face without eyewear (domain $A$) to a face with glasses (domain $B$). While most methods would map to a person with any glasses, our solution is guided and we attach to an image $\boldsymbol{a} \in A$, the glasses that are present in a reference image $\boldsymbol{b} \in B$.

In comparison to other guided image to image translation methods, our method is considerably simpler. It relies on having a latent space with two parts: (i) a shared part that is common to both $A$ and $B$, and (ii) a specific part that encodes the added content in $B$. By setting the second part to be the zero vector for all samples in $A$, a disentanglement emerges. Our analysis shows that this

Table 1: A comparison to other unsupervised guided image to image translation methods. $^{\dagger}k = 5$ is the number of pre-segmented face parts. $^{\ddagger}$Used for domain confusion, not on the output.

| | | MUNIT (Huang, 2018) | EG-UNIT (Ma, 2018) | DRIT (Lee, 2018) | PairedCy-cleGAN (Chang'18) | Our |
|---|---|---|---|---|---|---|
| **Sharing pattern** | Shared layers | | + | + | | |
| | Shared latent Space | + | | + | | + |
| | Shared encoder | | | | | + |
| | Shared decoder | | | | | + |
| **Number of networks** | Encoders | 4 | 4 | 4 | | 2 |
| | Generators | 2 | 2 | 2 | 2k$^{\dagger}$ | 1 |
| | Discriminators | 2 | 2 | 2 | k$^{\dagger}$ | 1$^{\ddagger}$ |
| | Other | | | 2 | | |

leads to the ability to train with a simple, straightforward domain confusion term, while enjoying the generalization guarantees that would otherwise require a more elaborate loss.

## 1.1 PREVIOUS WORK

In image to image translation, the transformations are often captured by multi-layered networks of an encoder-decoder architecture. The existing solutions often assume a one to one mapping between the domains, i.e., that there exists a function $y$ such that given a sample $a$ in domain $A$, maps it to an analog sample in domain $B$. In fact, the circularity based constraints by Zhu et al. (2017a); Kim et al. (2017); Yi et al. (2017) are based on this assumption, since going from one domain to the other and back, it is assumed that the original sample is obtained, which requires no loss of information. However, to employ an example made popular by Zhu et al. (2017a), when going from a zebra to a horse, the stripes are lost, which results in an ambiguity when mapping in the other direction.

This shortcoming was identified by the literature, and a few contributions present many to many mappings. These include the augmented CycleGAN Almahairi et al. (2018), which adds a random vector to each domain. A completely different approach is taken by the NAM method (Hoshen & Wolf, 2018), in which multiple solutions are obtained by considering multiple random initializations. In our work, multiplicity of outcomes arises from using a reference (guide) image.

A powerful way to capture relations between the two domains, is by employing separate autoencoders for the two domains, but which share many of the weights (Liu & Tuzel, 2016; Liu et al., 2017). This leads to a shared representation: while the low-level image properties, such as texture and color, are domain-specific and encoded/decoded separately, the mid- and top-level properties are common to both domains and are processed by identical replicas of the same layers. In our work, we rely on a shared encoder for both domains in order to enforce a shared representation.

Our method performs one-sided mapping, from $A$ to $B$, and does not learn the mapping in the other direction. Benaim & Wolf (2017) perform one-sided mapping using a distance based constraint, which we do not employ. That method is symmetric in the two domains and in many experiments, the distance constraint is used in tandem with the cycle constraint. In our case, the two domains are asymmetric and one domain contains an added content. Our method is inherently asymmetric, reflecting the asymmetry between the source image and the guide image, which contains the additional content. The work by Taigman et al. (2017); Hoshen & Wolf (2018) map in an asymmetric way, but rely on the existence of a perceptual distance, which we do not employ.

**Guided Translation** The most relevant work contains very recent and concurrent methods in which the mapping between the domains employs two inputs: a source image $a$ and a reference (guide or attribute) image $b$. Tab. 1 compares these methods to ours along two axes: (i) where sharing occurs in the architecture, and (ii) the number of trained sub-networks of each type. The sharing can be of layers between different encoders and decoders, following, e.g., Liu & Tuzel (2016); sharing of a common part of a latent space; and using the same encoder or decoder for multiple

domains. The networks are of four types: encoders, which map images to a latent space, generators (also known as decoders), which generate images from a latent representation, discriminators that are used as part of an adversarial loss, and other, less-standard, networks.

It is apparent that our method is considerably simpler than the literature methods. The main reason is that our method is based on the emergence of disentanglement, as detailed in Sec. 4. This allows us to to train with many less parameters and without the need to apply excessive tuning, in order to balance or calibrate the various components of the compound loss.

The MUNIT architecture by Huang et al. (2018), like our architecture, employs a shared latent space, in addition to a domain specific latent space. Their architecture is not limited to two domains[1] and unlike ours, employs separate encoders and decoders for the various domains. The type of guiding that is obtained from the target domain in MUNIT is referred to as style, while in our case, the guidance provides content. Therefore, MUNIT, as can be seen in our experiments, cannot add specific glasses, when shifting from the no-glasses domain to the faces with eyewear domain.

The EG-UNIT architecture by Ma et al. (2018) presents a few novelties, including an adaptive method of masking-out a varying set of the features in the shared latent space. In our latent representation of domain $A$, some of the features are constantly zero, which is much simpler. This method also focuses on guiding for style and not for content, as is apparent form their experiments.

The very recent DRIT work by Lee et al. (2018) learns to map between two domains using a disentangled representation. Unlike our work, this work seems to focus on style rather than content. The proposed solution differs from us in many ways: (1) it relies on two-way mapping, while we only map from $A$ to $B$. (2) it relies on shared weights in order to ensure that the common representation is shared. (3) it adds a VAE-like (Kingma & Welling, 2014) statistical characterization of the latent space, which results in the ability to sample random attributes. As can be seen in Tab. 1, the solution of Lee et al. (2018) is considerably more involved than our solution.

DRIT (and also MUNIT) employ two different types of encoders that enforce a separation of the latent space representations to either style or content vectors. For example, the style encoder, unlike the content encoder, employs spatial pooling and it also results in a smaller representation than the content one. This is important, in the context of these methods, in order to ensure that the two representations encode different aspects of the image. If DRIT or MUNIT were to use the same type of encoder twice, then one encoder could capture all the information, and the image-based guiding (mixing representations from two images) would become mute. In contrast, our method (i) does not separate style and content, and (ii) has a representation that is geared toward capturing the additional content.

The work most similar to us in its goal, but not in method, is the PairedCycleGAN work by Chang et al. (2018). This work explores the single application of applying the makeup of a reference face to a source face image. Unfortunately, the method was only demonstrated on a proprietary unshared dataset and the code is also not publicly available, making a direct comparison impossible at this time. The method itself is completely different from ours and does not employ disentanglement. Instead, a generator with two image inputs is used to produce an output image, where the makeup is transfered between the input images, and a second generator is trained to remove makeup. The generation is done separately to $k = 5$ pre-segmented facial regions, and the generators do not employ an encoder-decoder architecture.

Lastly, there are guided methods, which are trained in the supervised domain, i.e., when there are matches between domain $A$ and $B$. Unlike the earlier one-to-one work, such as pix2pix Isola et al. (2017b), these methods produce multiple outputs based on a reference image in the target domain. Examples include the Bicycle GAN by Zhu et al. (2017b), who also applied, as baseline in their experiments, the methods of Bao et al. (2017); Gonzalez-Garcia et al. (2018).

**Other Disentanglement Work** InfoGAN (Chen et al., 2016) learns a representation in which, due to the statistical properties of the representations, specific classes are encoded as a one-hot encoding of part of the latent vector. In the work of Lample et al. (2017); Hadad et al. (2018), the representation is disentangled by reducing the class based information within it. The separate class based information is different in nature from our multi-dimensional added content. Cao et al. (2018),

---

[1]Ours method can be readily extended to multiple target domains $B_1, \ldots, B_k$, but this is not explored here.

which builds upon Hadad et al. (2018), performs guided image to image translation, but assumes the availability of class based information, which we do not.

## 2 PROBLEM SETUP

We consider a setting with two domains $A = (\mathbb{X}_A, D_A)$ and $B = (\mathbb{X}_B, D_B)$. Here, $\mathbb{X}_A, \mathbb{X}_B \subset \mathbb{R}^M$ and $D_A, D_B$ are distributions over them (resp.). The algorithm is provided with two independent datasets $\mathbb{S}_A = \{\boldsymbol{a}^i\}_{i=1}^{m_1}$ and $\mathbb{S}_B = \{\boldsymbol{b}^j\}_{j=1}^{m_2}$ of samples from the two domains that were sampled in the following manner:

$$\mathbb{S}_A \overset{\text{i.i.d}}{\sim} D_A^{m_1} \text{ and } \mathbb{S}_B \overset{\text{i.i.d}}{\sim} D_B^{m_2} \tag{1}$$

We denote, $D_{A,B} := D_A \times D_B$ the distribution of sampling $(\boldsymbol{a}, \boldsymbol{b})$ for $\boldsymbol{a} \sim D_A$ and $\boldsymbol{b} \sim D_B$ independently. We assume a generative model, in which $\boldsymbol{b}$ is specified by a sample $\boldsymbol{a}$ and a specification $\boldsymbol{c}$ from a third unknown domain $C = (\mathbb{X}_C, D_C)$ of *specifications* where $D_C$ is a distribution over the metric space $\mathbb{X}_C \subset \mathbb{R}^N$. Formally, there is an invertible function $u(\boldsymbol{b}) = (u_1(\boldsymbol{b}), u_2(\boldsymbol{b})) \in \mathbb{X}_A \times \mathbb{X}_C$ that takes a sample $\boldsymbol{b} \in \mathbb{X}_B$ and returns the content $u_1(\boldsymbol{b})$ of $\boldsymbol{b}$ and the specification $u_2(\boldsymbol{b})$ of $\boldsymbol{b}$. The goal is to learn a target function $y : \mathbb{X}_A \times \mathbb{X}_B \to \mathbb{X}_B$ such that:

$$y(\boldsymbol{a}, \boldsymbol{b}) \sim D_B \text{ where: } \boldsymbol{a} \sim D_A, \boldsymbol{b} \sim D_B, \boldsymbol{a} \perp\!\!\!\perp \boldsymbol{b} \text{ and } u(y(\boldsymbol{a}, \boldsymbol{b})) = (\boldsymbol{a}, u_2(\boldsymbol{b})) \tag{2}$$

Informally, the function $y$ takes two samples $\boldsymbol{a}$ and $\boldsymbol{b}$ and returns the analog of $\boldsymbol{a}$ in $B$ that has the specification of $\boldsymbol{b}$. For example, $A$ is the domain of images of persons, $B$ is the domain of images of persons with sunglasses and $C$ is the domain of images of sunglasses. The function $y$ takes an image of a person and an image of a person with sunglasses and returns an image of the first person with the specified sunglasses. For simplicity, we assume that the target function is extended to inputs $(\boldsymbol{b}_1, \boldsymbol{b}_2) \in \mathbb{X}_B^2$ and $u(y(\boldsymbol{b}_1, \boldsymbol{b}_2)) = (u_1(\boldsymbol{b}_1), u_2(\boldsymbol{b}_2))$. In other words, $\boldsymbol{b}_1$ and $\boldsymbol{b}_2$ are mapped to a third $\boldsymbol{b}$ that has the content of $\boldsymbol{b}_1$ and the specification of $\boldsymbol{b}_2$. In particular, $u(y(\boldsymbol{b}, \boldsymbol{b})) = (u_1(\boldsymbol{b}), u_2(\boldsymbol{b})) = u(\boldsymbol{b})$ and, therefore, $y(\boldsymbol{b}, \boldsymbol{b}) = \boldsymbol{b}$.

Note that within Eq. 2, there is an assumption on the underlying distributions $D_A$ and $D_B$. Using the concrete example, our framework assumes that the distribution of images of persons with sunglasses and the distribution of images of persons without them is the same, except for the sunglasses. Otherwise, the distribution of the samples generated by $y$ when resampling $\boldsymbol{a}$ would not be the same as $D_B$. Note that we do not enforce this assumption on the data, and only employ it for our theoretical results, to avoid additional terms.

For two functions $f_1, f_2 : \mathbb{X} \to \mathbb{R}$ and a distribution $D$ over $\mathbb{X}$, we define the generalization risk between $f_1$ and $f_2$ as follows:

$$R_D[f_1, f_2] := \mathbb{E}_{\boldsymbol{x} \sim D}[\ell(f_1(\boldsymbol{x}), f_2(\boldsymbol{x}))] \tag{3}$$

For a loss function $\ell : \mathbb{R}^M \times \mathbb{R}^M \to [0, \infty)$. Typically, we use the $L_1(\boldsymbol{u}, \boldsymbol{v}) := \|\boldsymbol{u} - \boldsymbol{v}\|_1$ or $L_2(\boldsymbol{u}, \boldsymbol{v}) := \|\boldsymbol{u} - \boldsymbol{v}\|_2^2$ losses. The goal of the algorithm is to return a hypothesis $h \in \mathcal{H}$, such that $h : \mathbb{R}^M \times \mathbb{R}^M \to \mathbb{R}^M$, that minimizes the generalization risk,

$$R_{D_{A,B}}[h, y] = \mathbb{E}_{(\boldsymbol{a}, \boldsymbol{b}) \sim D_{A,B}}[\ell(h(\boldsymbol{a}, \boldsymbol{b}), y(\boldsymbol{a}, \boldsymbol{b}))] \tag{4}$$

This quantity measures the expected loss of $h$ in mapping two samples $\boldsymbol{a} \sim D_A$ and $\boldsymbol{b} \sim D_B$ to the analog $y(\boldsymbol{a}, \boldsymbol{b})$ of $\boldsymbol{a}$, that has the specification $u_2(\boldsymbol{b})$ of $\boldsymbol{b}$. The main challenge is that the algorithm does not observes paired examples of the form $((\boldsymbol{a}, \boldsymbol{b}), y(\boldsymbol{a}, \boldsymbol{b}))$ as a direct supervision for learning the mapping $y : \mathbb{X}_A \times \mathbb{X}_B \to \mathbb{X}_B$.

## 3 METHOD

In order to learn the mapping $y$, we only use an encoder-decoder architecture, in which the encoder receives two input samples and the decoder produces a single output sample that borrows from both input samples. As we discuss in Sec. 2, the goal of the algorithm is to learn a mapping $h = g \circ f \in \mathcal{H}$ such that: $g \circ f(\boldsymbol{a}, \boldsymbol{b}) \approx y(\boldsymbol{a}, \boldsymbol{b})$. Here, $f$ serves as an encoder and $g$ as a decoder. The encoder $f$ in our framework is a member of a set of encoders $\mathcal{F}$, each decomposable into two parts and takes the following form:

$$f(\boldsymbol{a}, \boldsymbol{b}) = (e_1(\boldsymbol{a}), e_2(\boldsymbol{b})) \tag{5}$$

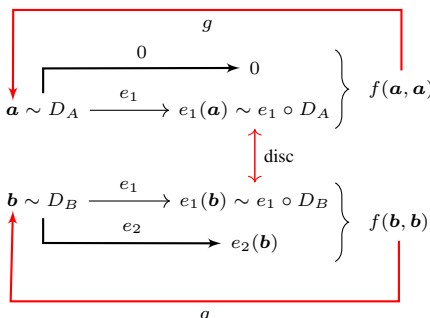

Figure 1: An illustration of the domains and functions employed in our work. $D_A$ and $D_B$ are the distributions of images in the two domains. $e_1$ and $e_2$ the two pathways of the encoder, $g$ is the decoder, which is applied to $f$, which aggregates the output of the two encoder pathways. There are only three constraints used while training, shown in red: (i) a reconstruction loss in domain $A$, comparing $g \circ f(\boldsymbol{a}, \boldsymbol{a})$ with a, (ii) a reconstruction loss in domain $B$, and (iii) a measure of the discrepancy between the distributions $e_1 \circ D_A$ and $e_1 \circ D_B$, measured with a domain confusion term.

where $e_1 : \mathbb{R}^M \to \mathbb{R}^{E_1}$ serves as an encoder of shared content and $e_2 : \mathbb{R}^M \to \mathbb{R}^{E_2}$ serves as an encoder of specific content. Here, $E_1$ and $E_2$ are the dimensions of the encodings. The decoder, $g$ is a member of a set of decoders $\mathcal{M}$. Each member of $\mathcal{M}$ is a function $g : \mathbb{R}^{E_1+E_2} \to \mathbb{R}^M$. In order to learn the functions $f$ and $g$, we apply the following min-max optimization:

$$\min_{f \in \mathcal{F}, g \in \mathcal{M}} \max_{d \in \mathcal{C}} \{\mathcal{L}_A + \mathcal{L}_B - \lambda \mathcal{L}_D\}, \tag{6}$$

for some weight parameter $\lambda > 0$, of the following training losses (see Fig. 1):

$$\mathcal{L}_A = \frac{1}{m_1} \sum_{\boldsymbol{a} \in \mathbb{S}_A} \|g(e_1(\boldsymbol{a}), 0_{E_2}) - \boldsymbol{a}\|_1 \tag{7}$$

$$\mathcal{L}_B = \frac{1}{m_2} \sum_{\boldsymbol{a} \in \mathbb{S}_B} \|g(e_1(\boldsymbol{b}), e_2(\boldsymbol{b})) - \boldsymbol{b}\|_1 \tag{8}$$

$$\mathcal{L}_D = \frac{1}{m_1} \sum_{\boldsymbol{a} \in \mathbb{S}_A} l(d(e_1(\boldsymbol{a})), 0) + \frac{1}{m_2} \sum_{\boldsymbol{b} \in \mathbb{S}_B} l(d(e_1(\boldsymbol{b})), 1) \tag{9}$$

where $0_{E_2}$ is the vector of zeros of length $E_2$, $d$ is a discriminator network, and $l(p, q) = -(q \log(p) + (1-q) \log(1-p))$ is the binary cross entropy loss for $p \in [0, 1]$ and $q \in \{0, 1\}$. The discriminator $d$ is a member of a set of discriminators $\mathcal{C}$ that locates functions $d : \mathbb{R}^M \to [0, 1]$.

The discriminator $d$ is trained to minimize $\mathcal{L}_D$ and Eq. 9 is a domain confusion term (Ganin et al., 2016) encouraging the distribution $e_1 \circ D_A$ to be similar to the distribution $e_1 \circ D_B$. Here and elsewhere, the composition $f \circ D$ of a function $f$ and a distribution $D$ denotes the distribution of $f(\boldsymbol{x})$ for $\boldsymbol{x} \sim D$.

## 4 ANALYSIS

In this section, we provide a theoretical analysis for the success of the proposed method. For this purpose, we recall a few technical notations (Cover & Thomas, 2006): the expectation and probability operators symbols $\mathbb{E}$, $\mathbb{P}$, the Shannon entropy (discrete or continuous) $H(X) := -\mathbb{E}_X[\log \mathbb{P}[X]]$, the conditional entropy $H(X|Y) := H(X, Y) - H(Y)$, the (conditional) mutual information (discrete or continuous) $I(X; Y|Z) := H(X|Z) - H(X|Y, Z)$, the Kullback-Leibler (KL) divergence $D_{\mathrm{KL}}(p\|q) := \mathbb{E}_{\boldsymbol{x} \sim p}[\log(p(\boldsymbol{x})/q(\boldsymbol{x}))]$, and the total correlation $TC(\boldsymbol{z}) := D_{\mathrm{KL}}(\mathbb{P}[\boldsymbol{z}] \| \prod_i \mathbb{P}[\boldsymbol{z}_i])$, where $\mathbb{P}[\boldsymbol{z}_i]$ is the marginal distribution of the $i$'th component of $\boldsymbol{z}$. In particular, $TC(\boldsymbol{z})$ is zero if and only if the components of $\boldsymbol{z}$ are independent, in which case we say that $\boldsymbol{z}$ is disentangled. For two distributions $D_1$ and $D_2$, we define the $\mathcal{C}$-discrepancy between them to be $\mathrm{disc}_{\mathcal{C}}(D_1, D_2) := \sup_{c_1, c_2 \in \mathcal{C}} |R_{D_1}[c_1, c_2] - R_{D_2}[c_1, c_2]| = \sup_{c_1, c_2} |\mathbb{E}_{x \sim D_1} \ell(c_1(\boldsymbol{x}), c_2(\boldsymbol{x})) - \mathbb{E}_{x \sim D_2} \ell(c_1(\boldsymbol{x}), c_2(\boldsymbol{x}))|$. The discrepancy behaves as an adversarial distance measure between two distributions, where $d(\boldsymbol{x}) = \ell(c_1(\boldsymbol{x}), c_2(\boldsymbol{x}))$ is the discriminator that tries to differentiate between $D_1$ and $D_2$, for $c_1, c_2 \in \mathcal{C}$. This quantity is being employed in Chazelle (2000); Ben-david et al. (2006); Mansour et al. (2009); Cortes & Mohri (2014).

## 4.1 GENERALIZATION BOUND

Thm. 1 upper bounds the generalization risk, based on terms that can be minimized during training, as well as on approximation terms. It is similar in fashion to the classic domain adaptation bounds proposed by Ben-David et al. (2010); Mansour et al. (2009).

**Theorem 1.** *Assume that the loss function $\ell$ is symmetric and obeys the triangle inequality. Then, for any autoencoder $h = g \circ f \in \mathcal{H}$, such that $f(\boldsymbol{x}_1, \boldsymbol{x}_2) = (e_1(\boldsymbol{x}_1), e_2(\boldsymbol{x}_2)) \in \mathcal{F}$ is an encoder and $g \in \mathcal{M}$ is a decoder, the following holds,*

$$R_{D_{A,B}}[h, y] \leq R_{D_{B,\hat{B}}}[h, y] + \min_{g^* \in \mathcal{M}} \left\{ R_{D_{A,B}}[g^* \circ f, y] + R_{D_{B,\hat{B}}}[g^* \circ f, y] \right\} \tag{10}$$
$$+ \operatorname{disc}_{\mathcal{M}}(f \circ D_{A,B}, f \circ D_{B,\hat{B}})$$

*where $D_{B,\hat{B}}$ is the distribution of $(\boldsymbol{b}, \boldsymbol{b})$ where $\boldsymbol{b} \sim D_B$.*

(The proofs can be found in the appendix.) Thm. 1 provides an upper bound on the generalization risk $R_{D_{A,B}}[h, y]$, which is the argument that we would like to minimize. The upper bound is decomposed of three terms: a reconstruction error, an approximation error and a discrepancy term. The first term,

$$R_{D_{B,\hat{B}}}[h, y] = \mathbb{E}_{(\boldsymbol{b},\boldsymbol{b}) \sim D_{B,\hat{B}}}[\ell(g \circ f(\boldsymbol{b}, \boldsymbol{b}), y(\boldsymbol{b}, \boldsymbol{b}))] = \mathbb{E}_{\boldsymbol{b} \sim D_B}[\ell(g \circ f(\boldsymbol{b}, \boldsymbol{b}), \boldsymbol{b})] \tag{11}$$

is the reconstruction error for samples $\boldsymbol{b} \sim D_B$. Since we do not have full access to $D_B$, we minimize its empirical version (see Eq. 8). The second term, $\min_{g \in \mathcal{M}} \left\{ R_{D_{A,B}}[g \circ f, y] + R_{D_{B,\hat{B}}}[g \circ f, y] \right\}$, measures the minimal error obtained by a best fitting $g \in \mathcal{M}$, such that $g \circ f \approx y$ for inputs $(\boldsymbol{a}, \boldsymbol{b}) \sim D_{A,B}$ and for inputs $(\boldsymbol{b}, \boldsymbol{b}) \sim D_{B,\hat{B}}$. Similar to (Ben-David et al., 2010; Mansour et al., 2009), this term is assumed to be small and is decreased as $\mathcal{M}$'s capacity is increased. The third term, $\operatorname{disc}_{\mathcal{M}}(f \circ D_{A,B}, f \circ D_{B,\hat{B}})$, is the discrepancy between the distributions $f \circ D_{A,B}$ and $f \circ D_{B,\hat{B}}$. This term is small, if the distributions of $(e_1(\boldsymbol{a}), e_2(\boldsymbol{b}))$ (for $\boldsymbol{a} \sim D_A$ and $\boldsymbol{b} \sim D_B$ independently) and $(e_1(\boldsymbol{b}), e_2(\boldsymbol{b}))$ (for $\boldsymbol{b} \sim D_B$) are close to each other. Since $e_1(\boldsymbol{a})$ and $e_2(\boldsymbol{b})$ are independent of each other (from the factorization $D_{A,B} = D_A \times D_B$), if this term is small, then, $e_1(\boldsymbol{b})$ and $e_2(\boldsymbol{b})$ weakly depend on each other. Moreover, if the discrepancy term is zero, then, $e_1(\boldsymbol{b})$ and $e_2(\boldsymbol{b})$ are independent of each other.

While one can minimize the discrepancy term explicitly, by minimizing it with respect to $e_1$ and $e_2$, using a discriminator, we found empirically that this confusion term, which involves both parts of the embedding, is highly unstable. Instead, we show theoretically and empirically that there is a high likelihood for a disentangled representation (where $e_1(b)$ and $e_2(b)$ are independent) to emerge, and the discrepancy term can be replaced with the following discrepancy $\operatorname{disc}_{\mathcal{M}'}(e_1 \circ D_A, e_1 \circ D_B)$, which measures the closeness between the distributions of $e_1(\boldsymbol{a})$ and of $e_1(\boldsymbol{b})$ for $\boldsymbol{a} \sim D_A$ and $\boldsymbol{b} \sim D_B$, as is done in Eq. 9. Here, $\mathcal{M}'$ is a set of discriminators that are similar in complexity to the ones in $\mathcal{M}$. This discrepancy is simpler than the one in Eq. 10, since it does not involve a comparison of $e_2$ between two distributions, nor the interaction between $e_1$ and $e_2$.

In Lem. 1, we show that if $e_1(\boldsymbol{b})$ and $e_2(\boldsymbol{b})$ are independent, then, $\operatorname{disc}(f \circ D_{A,B}, f \circ D_{B,\hat{B}}) \leq \operatorname{disc}(e_1 \circ D_A, e_1 \circ D_B)$. Therefore, if a disentangled representation occurs, we can minimize $\operatorname{disc}(f \circ D_{A,B}, f \circ D_{B,\hat{B}})$, by minimizing $\operatorname{disc}(e_1 \circ D_A, e_1 \circ D_B)$ instead.

**Lemma 1.** *Let $\mathcal{M}$ be the set of neural networks of the form: $c(\boldsymbol{x}) = \phi(\boldsymbol{W}_r \dots \phi(\boldsymbol{W}_2 \phi(\boldsymbol{W}_1 \boldsymbol{x} + \boldsymbol{q})))$, where, $\boldsymbol{W}_i \in \mathbb{R}^{d_i \times d_{i+1}}$ for $i \in \{1, \dots, r - 1\}$, $\boldsymbol{q} \in \mathbb{R}^{d_2}$ and $d_1 = E_1 + E_2$. In addition, $\phi(x_1, \dots, x_k) = (\phi_1(x_1), \dots, \phi_1(x_k))$, for $k \in \mathbb{N}$, $(x_1, \dots, x_k) \in \mathbb{R}^k$ and a non-linear activation function $\phi_1 : \mathbb{R} \to \mathbb{R}$. Let $\mathcal{M}'$ be the same as $\mathcal{M}$ with $d_1 = E_1$ (instead of $d_1 = E_1 + E_2$). Let $f(\boldsymbol{x}) = (e_1(\boldsymbol{x}), e_2(\boldsymbol{x}))$ be an encoder and assume that: $e_1(\boldsymbol{b}) \perp\!\!\!\perp e_2(\boldsymbol{b})$. Then,*

$$\operatorname{disc}_{\mathcal{M}}(f \circ D_{A,B}, f \circ D_{B,\hat{B}}) \leq \operatorname{disc}_{\mathcal{M}'}(e_1 \circ D_A, e_1 \circ D_B) \tag{12}$$

## 4.2 EMERGENCE OF DISENTANGLED REPRESENTATIONS

The following results are very technical and inherit many of the assumptions used by previous work. We therefore state the results informally here and leave the complete exposition to the appendix.

First, we extend Proposition 5.2 of Achille & Soatto (2018) from the case of multiclass classification to the case of autoencoders. In their work, the aim is to show the conditions in which a mid-level representation $f$ of a multi-class classification neural network $h = c \circ f$ is both disentangled, i.e., $TC(f(\boldsymbol{b}))$ is small, and is minimal, i.e., $I(f(\boldsymbol{b}); \boldsymbol{b})$ is small, where $\boldsymbol{b}$ is an input random variable. In their Proposition 5.2, they focus on a linear representation, i.e., $f = \boldsymbol{W}$ is a linear transformation, and they introduce a tight upper bound on the sum $TC(\boldsymbol{W}\boldsymbol{b}) + I(\boldsymbol{W}\boldsymbol{b}; \boldsymbol{b})$.

In the general case, their goal is to show that for a neural network $h = c \circ f$, both quantities $TC(f(\boldsymbol{b}))$ and $I(f(\boldsymbol{b}); \boldsymbol{b})$ are small, when $f$ that is a high level representation of the input. Unfortunately, they were unable to show that both terms are small simultaneously. Therefore, in their Cor. 5.3, they extend the bound of their Proposition 5.2 to show that only the mutual information $I(f(\boldsymbol{b}); \boldsymbol{b})$ is small and assume that the components of each mid-level representation of $\boldsymbol{b}$ in the layers of $f$ are uncorrelated, which is a very restrictive assumption.

In our Lem. 2, we provide an upper bound for $TC(f(\boldsymbol{b}))$ that is similar in fashion to their bounds. The main differentiating factor is that we deal with an autoencoder $h = g \circ f$. In this case, the mutual information $I(h(\boldsymbol{b}); \boldsymbol{b})$ is expected to be large, since $h(\boldsymbol{b})$ is trained to recover $\boldsymbol{b}$ and by the data-processing inequality, $I(f(\boldsymbol{b}); \boldsymbol{b}) \geq I(h(\boldsymbol{b}); \boldsymbol{b})$ and therefore, $I(f(\boldsymbol{b}); \boldsymbol{b})$ is also large in this case. As a result, no upper bound is given on the mutual information term $I(f(\boldsymbol{b}); \boldsymbol{b})$. This is unlike the information bottleneck principle, which guides the classification setting of Achille & Soatto (2018).

Our upper bound on $TC(f(\boldsymbol{b}))$ includes the term $-I(h(\boldsymbol{b}); \boldsymbol{b})$, and consequently, $TC(f(\boldsymbol{b}))$ tends to be smaller (increased disentanglement) as $I(h(\boldsymbol{b}); \boldsymbol{b})$ increases. Additionally, the upper bound sums a term $d_1 \cdot q(\alpha)$. Here, $d_1$ is the dimension of $f(\boldsymbol{b})$, $\alpha$ denotes the amount of regularization in the weights of $f$ and $q(\alpha)$ is monotonically increasing as $\alpha$ tends to zero. Therefore, the disentanglement tends to be larger for an autoencoder $h = g \circ f$, such that $f$ is regularized and the mutual information $I(h(\boldsymbol{b}); \boldsymbol{b})$ is large. In our analysis, we do not require $f$ to be a linear transformation and we do not assume that the components of each mid-level representation of $\boldsymbol{b}$ in the layers of $f$ are uncorrelated.

**Lemma 2** (Informal). *Let $\boldsymbol{b} \sim D_B$ be a distribution and $h = g \circ f$ an autoencoder. Let $d_1$ be the dimension of $f(\boldsymbol{b})$ and $d_2$ the dimension of the layer previous to $f(\boldsymbol{b})$. Under some assumptions on the weights of the encoder, there is a monotonically decreasing function $q(\alpha)$ for $\alpha > 0$ such that:*

$$TC(f(\boldsymbol{b})) \leq d_1 \cdot q(\alpha) - I(h(\boldsymbol{b}); \boldsymbol{b}) + \mathcal{O}\left(d_1/d_2\right) \tag{13}$$

Eq. 13 bounds the total correlation of $f(\boldsymbol{b})$, which measures the amount of dependence between the components of the encoder on samples in $B$. The bounds has three terms: $d_1 \cdot q(\alpha)$, $-I(h(\boldsymbol{b}), \boldsymbol{b})$ and $\mathcal{O}\left(d_1/d_2\right)$. In this formulation, $\alpha$ denotes the amount of regularization in the weights of $f$. In addition, $q(\alpha)$ is monotonically increasing as $\alpha$ tends to zero. The term $I(h(\boldsymbol{b}); \boldsymbol{b})$ measures the mutual information between the input $\boldsymbol{b}$ and output $h(\boldsymbol{b})$ of the autoencoder $h$. Since the mutual information is subtracted in the right hand side, the larger it is, the smaller $TC(f(\boldsymbol{b}))$ should be. The last term, $\mathcal{O}(d_1/d_2)$ measures the ratio between the dimension of the output of $f$ and the dimension of the previous layer of $f$. Thus, this quantity is small whenever there is a significance reduction in the dimension in the application of the last layer of $f$.

Therefore, there is a tradeoff between the amount of regularization in the weights of $f$ and the mutual information $I(h(\boldsymbol{b}); \boldsymbol{b})$. If there is small regularization, then, the autoencoder is able to produce better reconstruction $h(\boldsymbol{b}) \approx \boldsymbol{b}$, and therefore, a larger value of $I(h(\boldsymbol{b}); \boldsymbol{b})$. On the other hand, small regularization leads to a higher value of $q(\alpha)$.

The bound relies on the mutual information between the inputs and outputs of the autoencoder to be large. The following lemma provides an argument why this is the case when the expected reconstruction error of the autoencoder is small.

**Lemma 3** (Informal). *Let $\boldsymbol{b} \sim D_B$ be a distribution over a discrete set $\mathbb{X}_B$ and $h = g \circ f$ an autoencoder. Assume that $\forall \boldsymbol{x}_1 \neq \boldsymbol{x}_2 \in \mathbb{X}_B : \|\boldsymbol{x}_1 - \boldsymbol{x}_2\|_1 > \Delta$. Then,*

$$I(h(\boldsymbol{b}); \boldsymbol{b}) \geq \left(1 - \frac{\mathbb{E}[\|h(\boldsymbol{b}) - \boldsymbol{b}\|_1]}{\Delta}\right) H(\boldsymbol{b}) - \sqrt{\mathbb{E}[\|h(\boldsymbol{b}) - \boldsymbol{b}\|_1]} \tag{14}$$

The above lemma asserts that if the samples in $D_B$ are well separated, whenever the autoencoder has a small expected reconstruction error, $\mathbb{E}[\|h(\boldsymbol{b}) - \boldsymbol{b}\|_1]$, then, the mutual information $I(h(\boldsymbol{b}); \boldsymbol{b})$ is at least a large portion of $H(\boldsymbol{b})$. Therefore, we conclude that if the autoencoder generalizes well, then, it also maximizes the mutual information $I(h(\boldsymbol{b}); \boldsymbol{b})$.

Table 2: Runtime and memory footprint statistics for our method, the two guided translation methods from the literature (MUNIT and DRIT), and the Fader network disentangled representation method.

| Measure | MUNIT | DRIT | Fader | Our |
|---|---|---|---|---|
| Performs guided mapping? | Yes | Yes | No | Yes |
| Number of iterations | 1.0M | 0.5M | 1.5M | 1.0M |
| Time per iteration | 0.78s | 1.05s | 0.14s | 0.15s |
| Memory footprint | 4.35 GB | 6.03 GB | 1.83 GB | 1.23 GB |
| Number of weighting parameters | 3 (require tuning) | 6 (require tuning) | 1 (fixed) | 1 (fixed) |

To conclude the analysis: for a small enough reconstruction error, when training the autoencoder, the mutual information between the autoencoder's input and output is high (Lem. 10), which implies that the individual coordinates of the representation layer are almost independent of each other (Lem. 9). When using part of the representation to encode the information that exists in domain $A$ (the shared part), the other part would contain coordinates that are weakly dependent of the features encoded in $A$. In such a case, we can train with a GAN that involves only the shared representation (Lem. 1). That way, we can upper bound the generalization error expressed in Thm. 1, using relatively simple loss terms, as is done in Sec. 3.

## 5 EXPERIMENTS

We evaluate our method on three additive facial attributes: eyewear, facial hair, and smile. Images from the celebA face image dataset by Yang et al. (2015) were used, since these are conveniently annotated as having the attribute or not. The images without the attribute (no glasses, or no facial-hair, or no smile) were used as domain $A$ in each of the experiments. Note that three different $A$ domains were used. As the second domain $B$, we used the images labeled as having glasses, having facial hair, or smiling, according to the experiment.

Our underlying network architecture adapts the architecture used by Lample et al. (2017), which is based on (Isola et al., 2017a), where we use Instance Normalization (Ulyanov et al., 2016) instead of Batch Normalization (Ioffe & Szegedy, 2015), and without dropout. Let $C_k$ denote a Convolution-InstanceNorm-ReLU layer with $k$ filters, where a kernel size of $4 \times 4$, with a stride of 2, and a padding of 1 is used. The activations of the encoders $e_1, e_2$ are leaky-ReLUs with a slope of 0.2 and the deocder $g$ employs ReLUs. $e_1$ has the following layers $C_{32}, C_{64}, C_{128}, C_{256}, C_{512}, C_{512-d}$; $e_2$ has a slightly lower capacity $C_{32}, C_{64}, C_{128}, C_{128}, C_{128}, C_d$, where $d = 25$. The input images have a size of $128 \times 128$, and the encoding is of size $512 \times 2 \times 2$ (split between the $e_1$ and $e_2$). $g$ is symmetric to the encoders and employs transposed convolutions for the upsampling.

In the first set of experiments, we add the relevant content from a random image $\boldsymbol{b} \in B$ into an image from $\boldsymbol{a}$. The results are given in Fig. 2, and appendix Fig. 7 and 8. We compare with two guided image translation baselines: MUNIT (Huang et al., 2018) and DRIT (Lee et al., 2018). We used the published code for each method and despite our best effort, these methods fail on the task of content addition. In almost all cases, the baseline methods apply the style of the guide and not the added content.

It should be noted that the simplicity of our approach directly translates to more efficient training than the two baselines methods. Our method has one weighting hyperparameter, which is fixed throughout the experiments. MUNIT and DRIT each has several weighting hyperparameters (since they use more loss terms) and these require attention and, need to change between the experiments, both in our runs and in the authors' own experiments. In addition, our method has a lower memory footprint and a much shorter duration of each iteration. The statistics are reported in Tab. 2 and as can be seen, the runtime and memory footprint of our method are much closer to the Fader network by Lample et al. (2017), which cannot perform guided mapping, than to MUNIT and DRIT.

Since the performance of the baselines is clearly inferior in the current setting, we did not hold a user study comparing different algorithms. Instead, we compare the output of our method directly with real images. Two experiments are conducted: (i) can users tell the difference between an image from domain $B$ and an image from domain $A$ that was translated to domain $B$, and (ii) can users tell

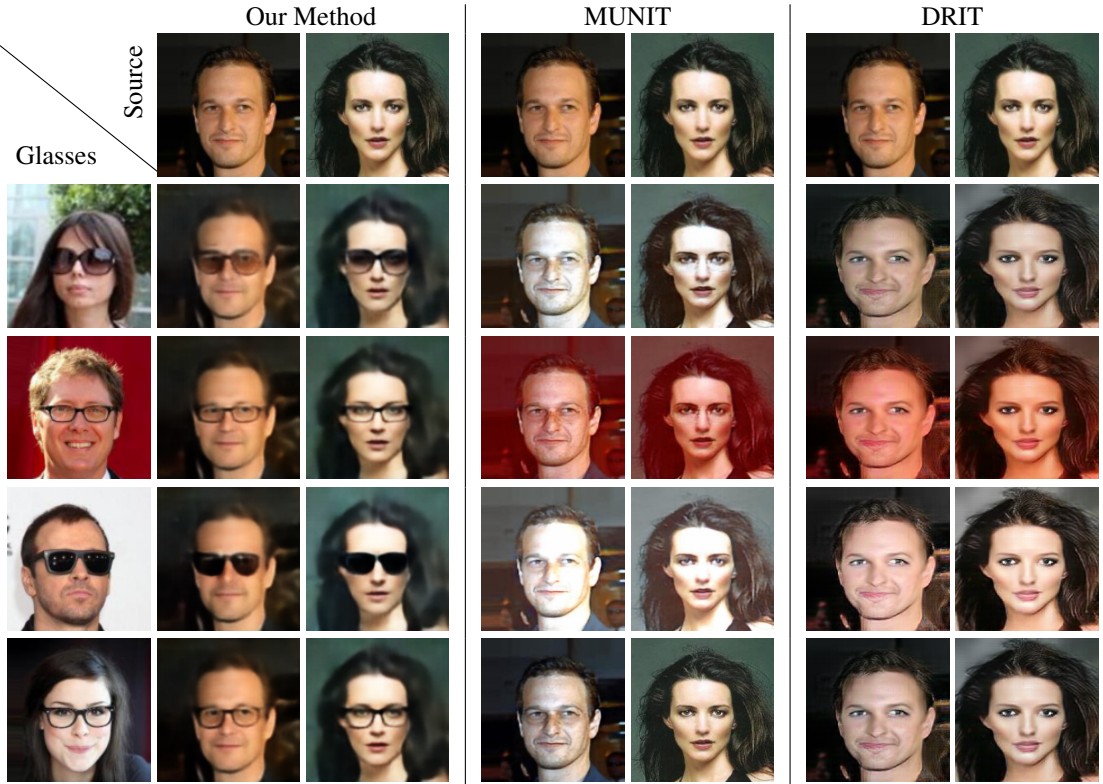

Figure 2: Glasses transfer. Our method vs literature baselines. Each image combines the domain $A$ image in the top row, with the content of the guide image on the left column.

the difference between an image from domain $B$ and the same image, after replacing the attribute's content (glasses, smile, or facial-hair) with that of another image from $B$. The experiment was performed with $n = 30$ users, who observed 10 pairs of images each, for each of the tests.

The results are reported in Tab. 3. As can be seen, users are able to detect the real image over the generated one, in most of the cases. However, the success ratio varies between the three image translation tasks and between the two types of comparisons. The most successful experiments, i.e., those where the users were confused the most, were in the facial hair ("beard") category. In contrast, when replacing a person's glasses with those of a random person, the users were able to tell the real image 74% of the time.

Fig. 3 and appendix Fig. 9 and 10 show the type of images shown in the experiment where users were asked to tell an image from domain $B$ from an hybrid image that contains a face of one image from this domain, and the attribute content from another image from it. As can be seen, most mix-and-match combinations seem natural. However, going over the rows, which should have a fixed attribute (e.g., the same glasses), one observes some variation. This unwanted variation arises from the need to fit the content to the new face.

The method does have, as can be expected, difficulty dealing with low quality inputs. Examples are shown in Fig. 4, including a misaligned source or guide image. Also shown is an example in which the added content in the target domain is very subtle. These challenges result in a lower quality output. However, the output in each case does indicate some ability to overcome the challenging input.

To evaluate the linearity of the latent representation $e_2(b)$, we performed interpolation experiments. The results are presented in Fig. 5. As can be seen, the change is gradual as we interpolate linearly between the $e_2$ encoding of the two guide images shown on the left and on the right.

In the supplementary appendix, we provide many more translation examples, see Fig. 11, 12, and 13.

Table 3: User study results. In each cell is the ratio of images, were users selected a real image as more natural than a generated one. Closer to 50% is better for the method.

| Forced choice performed by the user | Glasses | Smile | Facial Hair |
|---|---|---|---|
| Selected $\boldsymbol{b}$ over $g(e_1(\boldsymbol{a}), e_2(\boldsymbol{b'}))$, for $\boldsymbol{a} \in A, \boldsymbol{b}, \boldsymbol{b'} \in B$ | 58.2% | 63.4% | 51.7% |
| Selected $\boldsymbol{b}$ over $g(e_1(\boldsymbol{b}), e_2(\boldsymbol{b'}))$, for $\boldsymbol{b}, \boldsymbol{b'} \in B$ | 74.2% | 65.8% | 56.7% |

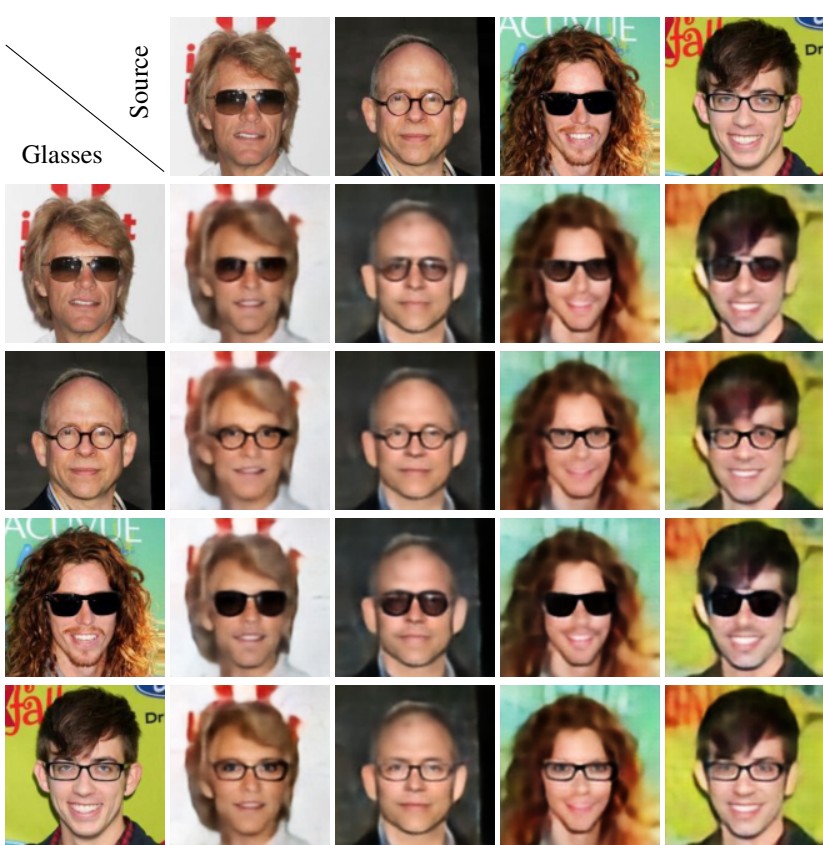

Figure 3: A mix and match experiment for glasses, using only domain $B$ images. Each image is a combination of the source image in the top row and the guide image on the left column.

While the previous experiments focused on the guided addition of content (mapping from $A$ to $B$), our method can also be applied in the other direction, from $B$ to $A$. This way, the specific content in image $\boldsymbol{b} \in B$ is removed. In our method, this is achieved simply by decoding a representation of the form $(e_1(\boldsymbol{b}), 0)$.

The advantage of mapping in this direction is the availability of additional literature methods to compare with, since no guiding is necessary. In Fig. 6, we compare the results we obtain for removing a feature with the Fader network method of Lample et al. (2017). As can be seen, the removal process of our method results in less residuals.

To verify that we obtain a better quality in comparison with that of the published implementation of Fader networks, we have applied both an automatic classifier and a user study. The classifier is trained on the training set of domain $A$ and $B$, using the same architecture that is used by Lample et al. (2017) to perform model selection.

Tab. 4 presents the mean probability of class $B$ provided by the classifier for the output of both Fader network and our method. As can be seen, the probability to belong to the class of the image before the transformation is, as desired, low for both methods. It is somewhat lower on average in our method, despite the fact that our method does not use such a network during training.

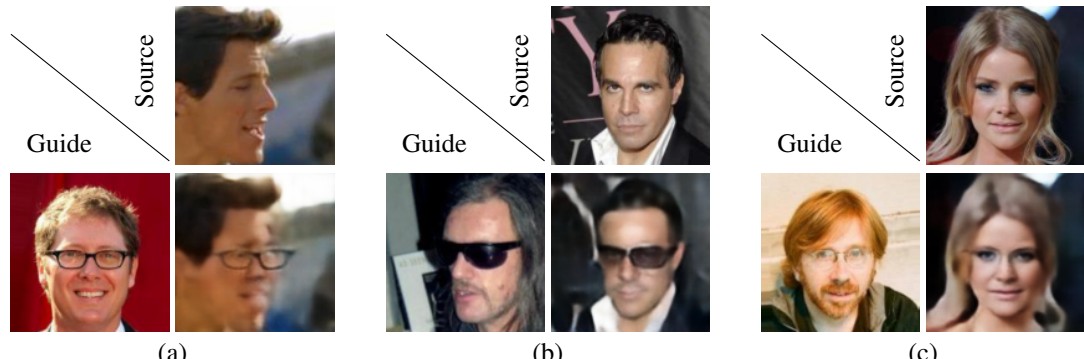

Figure 4: Some failure cases. (a) the source image is not well aligned. (b) the guide image is not well aligned. (c) the guide image has a very subtle content.

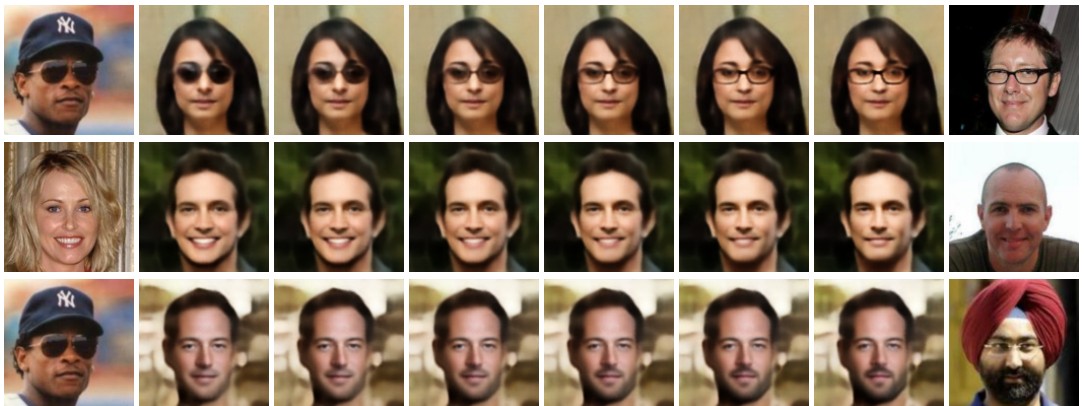

Figure 5: Interpolation experiments, where the content representation is linearly mixed between the one extracted from the left image and the one extracted from the right image.

The user study was conducted on $n = 20$ users, each examining 20 random test set triplets from each experiment. Each triplet showed the original image (with the feature) and the results of the two algorithms, where the feature is removed. The users preferred our method over fader 92% of the time for glasses removal, 89% of the time for facial hair removal, and 91% of the time for the removal of a smile.

## 6 CONCLUSIONS

When converting between two domains, there is an inherent ambiguity that arises from the domain-specific information in the target domain. In guided translation, the reference image in the target domain provides the missing information. Previous work has focused on the missing information that is highly tied to the texture of the image. For example, when translating between paintings and photos, DRIT adds considerable content from the reference photo. However, this is unstructured content, which is not well localized and is highly related to subsets of the image patches that exist in the target domain. In addition, the content from the reference photo that is out of the domain of paintings is not guaranteed to be fully present in the output.

Our work focuses on transformations in which the domain specific content is well structured, and guarantees to replicate all of the domain specific information from the reference image. This is done using a small number of networks and a surprisingly simple set of loss terms, which, due to the emergence of a disentangled representation, solves the problem convincingly.

Table 4: Classifier results for the image obtained after removing the desired feature. Results are the mean probability of domain $B$ for images that were transformed to domain $A$.

| Probability of class $B$ | Glasses | Smile | Facial Hair |
|---|---|---|---|
| Fader networks (Lample et al., 2017) | 0.066 | 0.064 | 0.182 |
| Our | 0.011 | 0.052 | 0.119 |

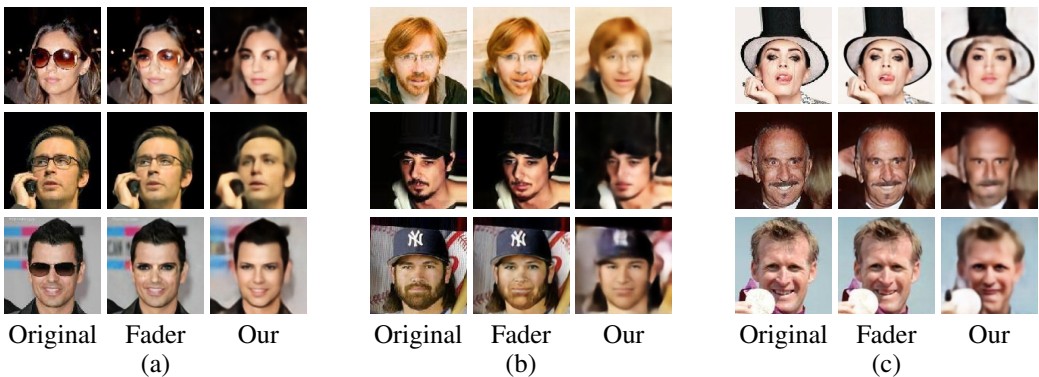

Original   Fader   Our       Original   Fader   Our       Original   Fader   Our

(a)              (b)           (c)

Figure 6: A comparison to the Fader networks of Lample et al. (2017) for the task of removing a feature. (a) Glasses. (b) Facial hair. (c) Mouth opening.

## ACKNOWLEDGEMENTS

This project has received funding from the European Research Council (ERC) under the European Unions Horizon 2020 research and innovation programme (grant ERC CoG 725974). The theoretical analysis in this work is part of Tomer Galanti's Ph.D thesis research conducted at Tel Aviv University.

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

# A ADDITIONAL FIGURES

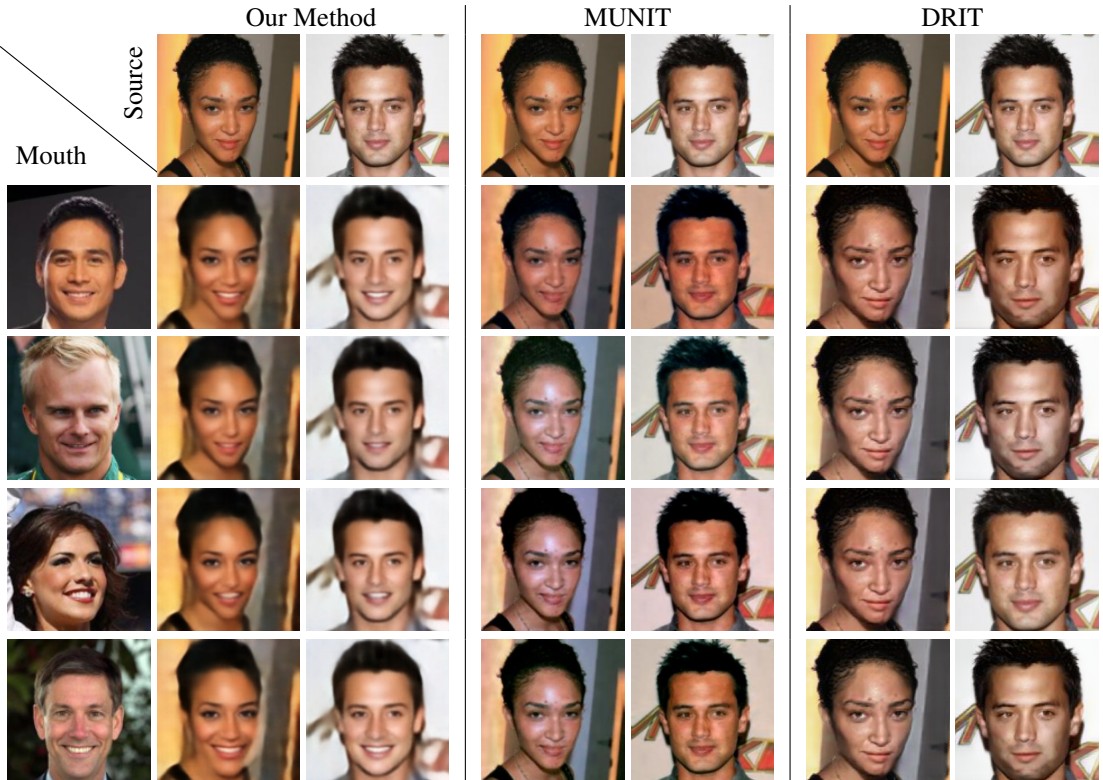

Figure 7: Smile transfer. Our method vs literature baselines.

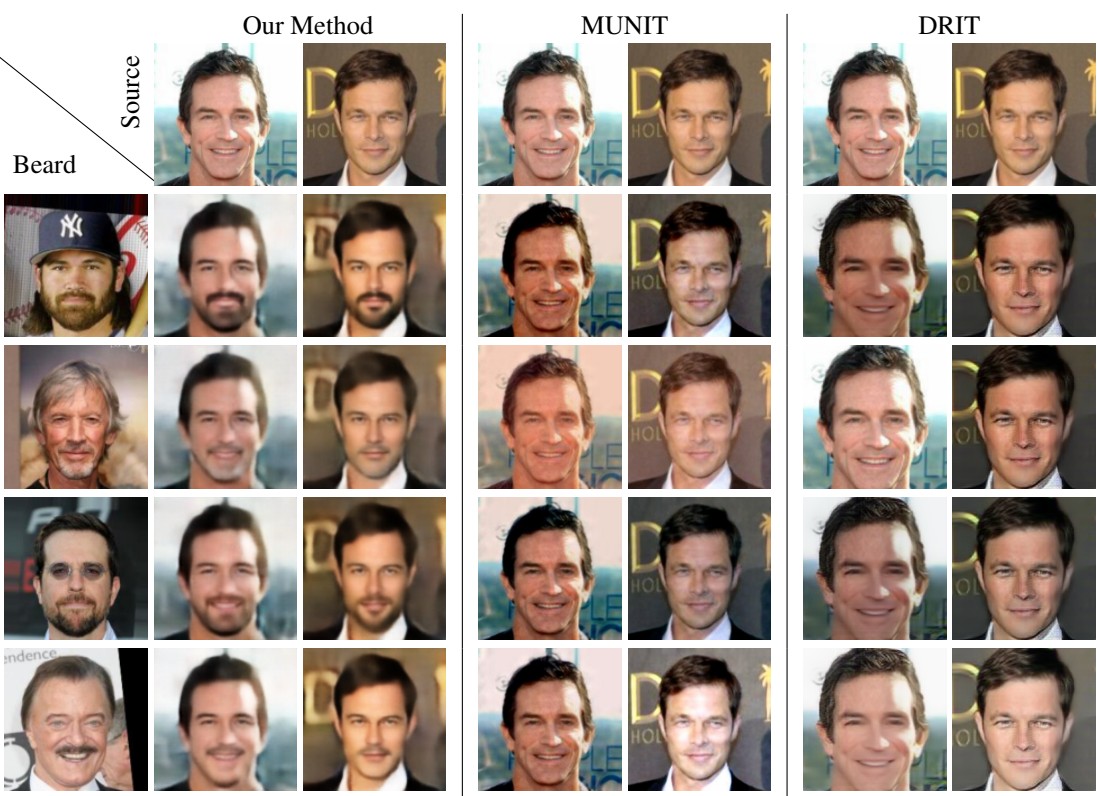

Figure 8: Facial hair transfer. Our method vs. the literature baselines.

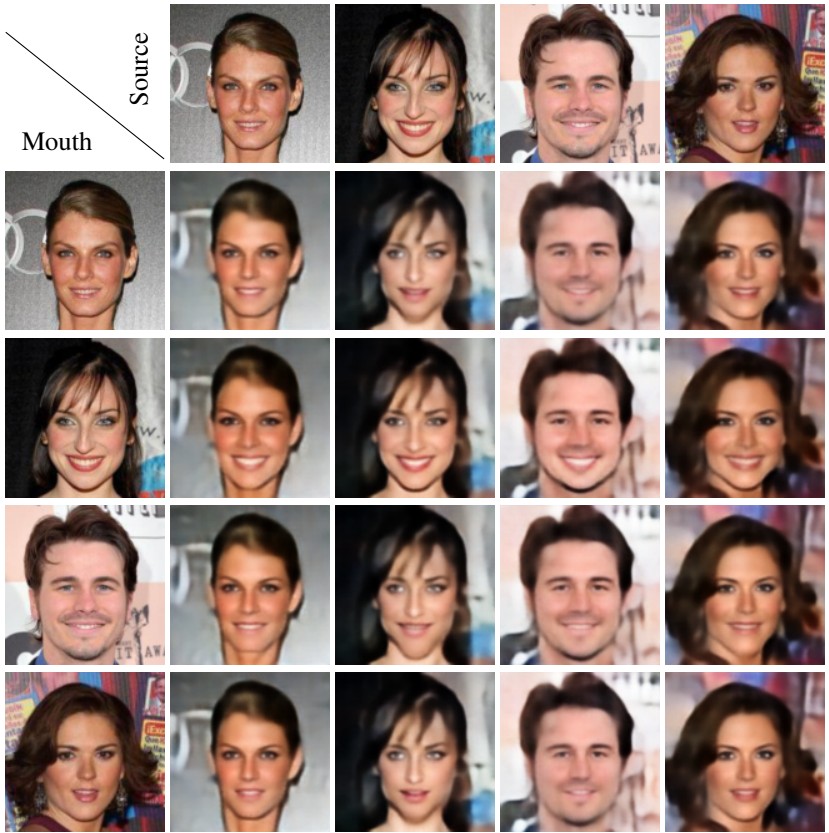

Figure 9: A mix and match experiment for no smile to smile translation, using only domain $B$ images.

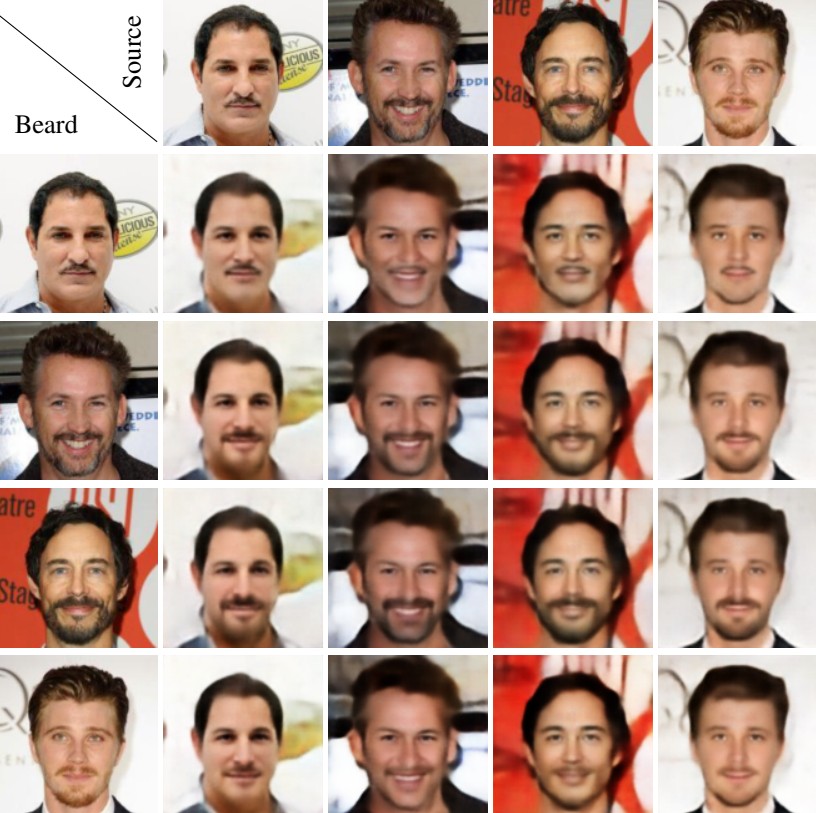

17

Figure 10: A mix and match experiment for the facial hair transfer, using only domain $B$ images.

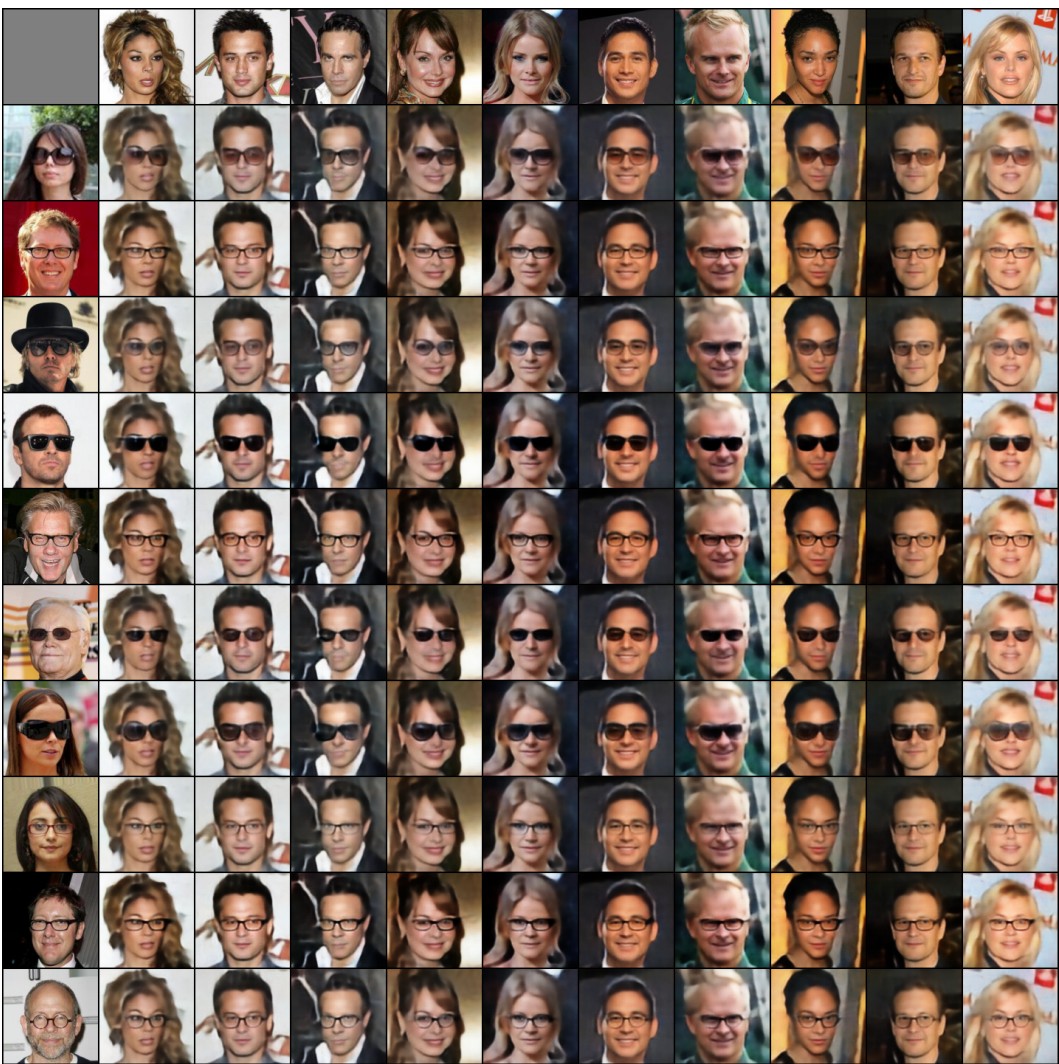

Figure 11: More eyewear transfer examples.

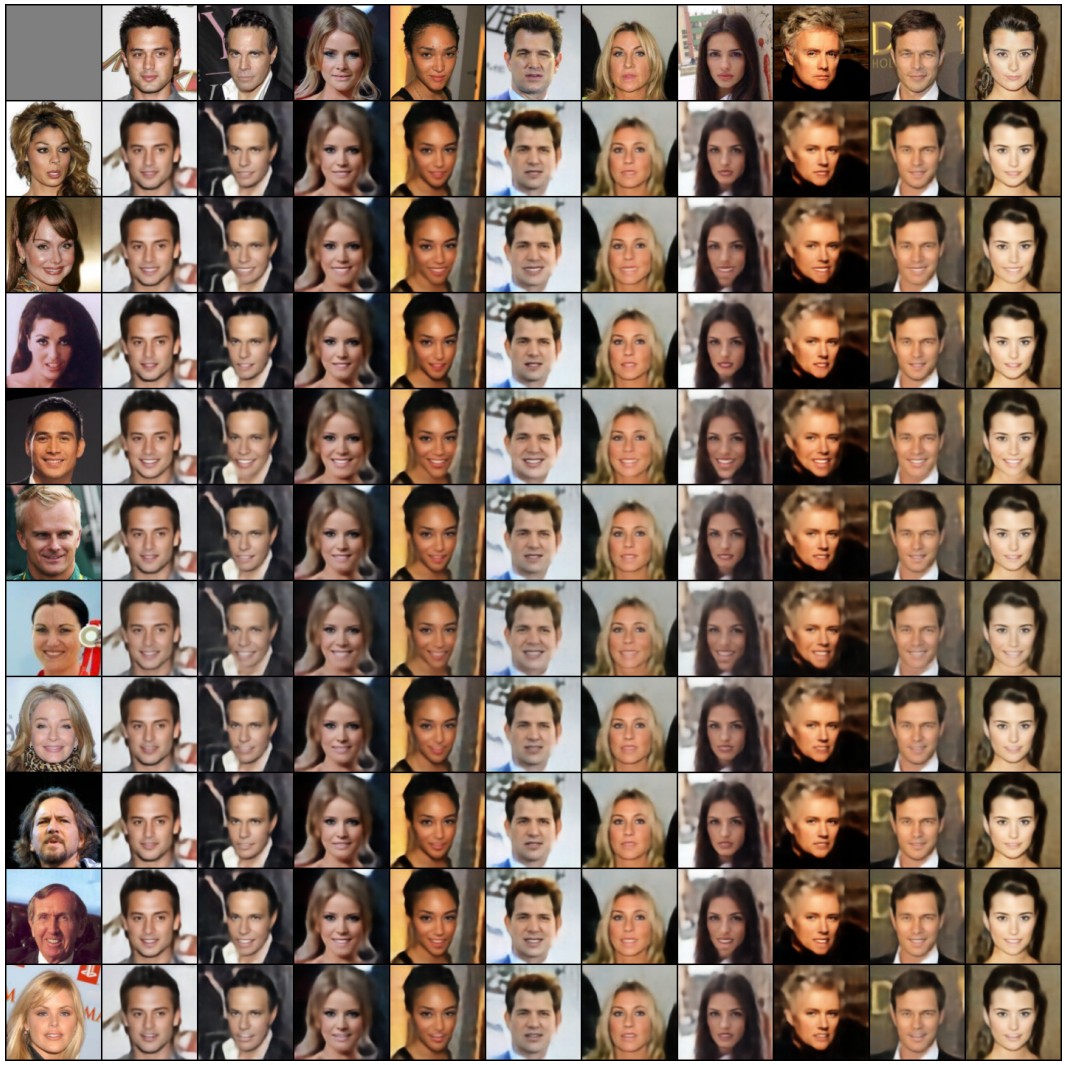

Figure 12: More smile transfer examples.

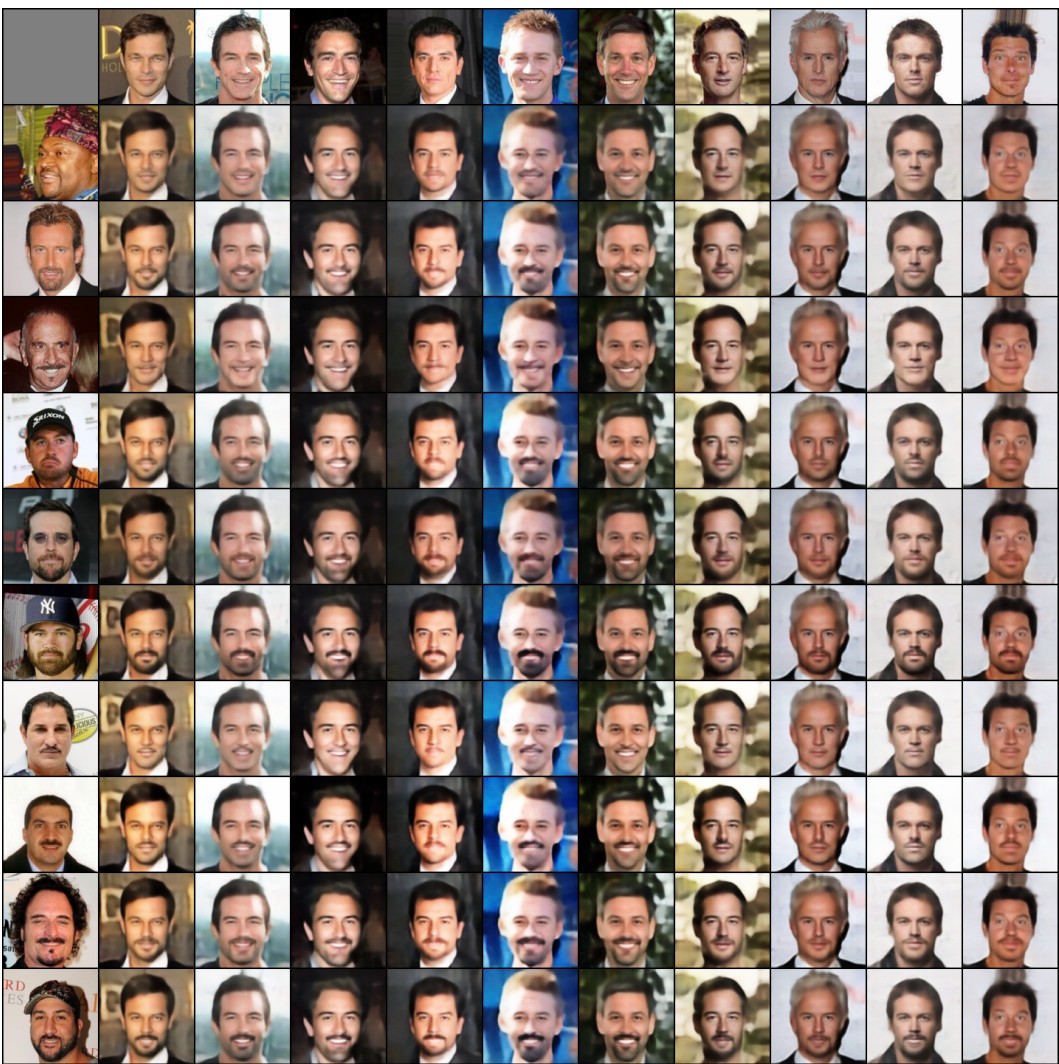

Figure 13: More facial hair transfer examples.

# B PRELIMINARIES

## B.1 NOTATIONS AND TERMINOLOGY

In this section we provide notations and terminology that are were not introduced in Sec. 4 but are necessary for the proofs of the claims in this section.

We say that three random variables (discrete or continuous) $X_1, X_2, X_3$ form a Markov chain, indicated with $X_1 \to X_2 \to X_3$, if $\mathbb{P}[X_3|X_2, X_1] = \mathbb{P}[X_3|X_2]$. The Data Processing Inequality (DPI) for a Markov chain $X_1 \to X_2 \to X_3$ ensures that $I(X_1; X_3) \leq \min(I(X_1; X_2), I(X_2; X_3))$. In particular, it holds for $X_2 = f(X_1)$ and $X_3 = g(X_2)$, where $f, g$ are deterministic processes.

We denote by $x \sim \log \mathcal{N}(\mu, \sigma^2)$ a random variable that is distributed by a log-normal distribution, i.e., $\log x \sim \mathcal{N}(\mu, \sigma^2)$. We consider that the mean and variance of a log-normal distribution $\log \mathcal{N}(\mu, \sigma^2)$ are $\exp(\mu + \sigma^2/2)$ and $(\exp(\sigma^2) - 1)\exp(2\mu + \sigma^2)$ respectively. We denote by $\boldsymbol{W} \odot \boldsymbol{U} := (\boldsymbol{W}_{k,j} \cdot \boldsymbol{U}_{k,j})_{k \leq m, j \leq m}$ the Hadamard product of two matrices $\boldsymbol{W}, \boldsymbol{U} \in \mathbb{R}^{m \times n}$. For a given vector $\boldsymbol{x} \in \mathbb{R}^m$, we denote $\dim(\boldsymbol{x}) := m$ and for a matrix $\boldsymbol{W} \in \mathbb{R}^{m \times n}$, we denote $\dim(\boldsymbol{W}) := mn$. In addition, we denote $\boldsymbol{x}^2 = \boldsymbol{x} \odot \boldsymbol{x} = (x_1^2, \ldots, x_m^2)$ and $\mathbb{E}[\boldsymbol{x}] = (\mathbb{E}[x_1], \ldots, \mathbb{E}[x_m])$. The indicator function, is denoted by $\mathbb{1}[x]$ for a boolean variable $x \in \{\text{true}, \text{false}\}$ (i.e., $\mathbb{1}[x] = 1$ if $x = \text{true}$ and $\mathbb{1}[x] = 0$ o.w).

## B.2 LEMMAS

In this section, we provide useful lemmas that aid in the proofs of our main results.

**Lemma 4.** *Let $\boldsymbol{x} = (x_1, \ldots, x_n) \in \mathbb{R}^n$ be a random vector. Let $\mu_1, \ldots, \mu_n : \mathbb{R} \to \mathbb{R}$ be continuous invertible functions and we denote $\mu(\boldsymbol{x}) := (\mu_1(x_1), \ldots, \mu_n(x_n))$. Then, $TC(\boldsymbol{x}) = TC(\mu(\boldsymbol{x}))$.*

*Proof.* First, we consider that:

$$TC(\boldsymbol{x}) = D_{\mathrm{KL}}\left(\mathbb{P}[\boldsymbol{x}] \middle\| \prod_{i=1}^n \mathbb{P}[x_i]\right) = D_{\mathrm{KL}}\left(\mathbb{P}[\boldsymbol{x}] \middle\| \mathbb{P}[\bar{\boldsymbol{x}}]\right) \tag{15}$$

where, $\bar{\boldsymbol{x}} := (\bar{x}_1, \ldots, \bar{x}_n)$ is a vector of independent random variables, such that $\bar{x}_i$ is distributed, according to the marginal distribution of $x_i$.

KL-divergence is invariant to applying continuous invertible transformations, i.e., $D_{\mathrm{KL}}(X\|Y) = D_{\mathrm{KL}}(\mu(X)\|\mu(Y))$ for $\mu$ that is continuous and invertible. Therefore,

$$TC(\boldsymbol{x}) = D_{\mathrm{KL}}\left(\mathbb{P}[\mu(\boldsymbol{x})] \middle\| \mathbb{P}[\mu(\bar{\boldsymbol{x}})]\right) = D_{\mathrm{KL}}\left(\mathbb{P}[\mu(\boldsymbol{x})] \middle\| \prod_{i=1}^n \mathbb{P}[\mu_i(\bar{x}_i)]\right) = TC(\mu(\boldsymbol{x})) \tag{16}$$

$\square$

**Lemma 5.** *Let $p \in [0, 1]$. Then, $H(p) \leq 2\log(2)\sqrt{p(1-p)}$.*

*Proof.* See (https://math.stackexchange.com/users/44121/jack daurizio).

The following lemma is a modification of Claim 2.1 in (Regev, 2013).

**Lemma 6.** *Let $X$ and $Y$ be two random variables. Assume that there is a function (i.e., a deterministic process) $F$, such that $\mathbb{P}[F(Y) = X] \geq q \geq 1/2$. Then, $I(X; Y) \geq qH(X) - H(q)$.*

*Proof.* By the data processing inequality,

$$
\begin{aligned}
I(X; Y) &\geq I(X; F(Y)) \\
&= H(X) - H(X|F(Y)) \\
&= H(X) - H(\mathbb{1}_{X=F(Y)}, X|F(Y)) \\
&= H(X) - (H(\mathbb{1}_{X=F(Y)}|F(Y)) + H(X|\mathbb{1}_{X=F(Y)}, F(Y)))
\end{aligned}
\tag{17}
$$

Since conditioning does not increase entropy, $H(\mathbb{1}_{X=F(Y)}|F(Y)) \leq H(\mathbb{1}_{X=F(Y)}) \leq H(q)$. In addition,

$$
\begin{aligned}
H(X|\mathbb{1}_{X=F(Y)}, F(Y)) =& \mathbb{P}[\mathbb{1}_{X=F(Y)} = 0] \cdot H(X|\mathbb{1}_{X=F(Y)} = 0, F(Y)) \\
& + \mathbb{P}[\mathbb{1}_{X=F(Y)} = 1] \cdot H(X|\mathbb{1}_{X=F(Y)} = 1, F(Y)) \\
=& \mathbb{P}[\mathbb{1}_{X=F(Y)} = 0] \cdot H(X|\mathbb{1}_{X=F(Y)} = 0, F(Y)) \\
\leq& (1-q)H(X|\mathbb{1}_{X=F(Y)} = 0, F(Y)) \\
\leq& (1-q)H(X)
\end{aligned}
\tag{18}
$$

Therefore, we conclude that, $I(X;Y) \geq qH(X) - H(q)$. $\qquad\square$

**Lemma 7.** *Let $\boldsymbol{b} \sim D_B$ be a distribution over a discrete set $\mathbb{X}_B \subset \mathbb{R}^M$ and $h_{\boldsymbol{v}} : \mathbb{R}^M \to \mathbb{R}^M$ is a (possibly random) function. Assume that $\forall \boldsymbol{x}_1 \neq \boldsymbol{x}_2 \in \mathbb{X}_B : \|\boldsymbol{x}_1 - \boldsymbol{x}_2\|_1 > \Delta$. Let $\mathbb{P}[\|h_{\boldsymbol{v}}(\boldsymbol{b}) - \boldsymbol{b}\|_1 \leq \Delta] \geq q \geq 1/2$. Then, $I(h_{\boldsymbol{v}}(\boldsymbol{b}); \boldsymbol{b}) \geq qH(\boldsymbol{b}) - H(q)$.*

*Proof.* Let $F(\boldsymbol{u}) := \arg\min_{\boldsymbol{x} \in \mathbb{X}_B} \|\boldsymbol{u} - \boldsymbol{x}\|_1$. Since the members of $\mathbb{X}_B$ are $\Delta$-distant from each other, if $\|h_{\boldsymbol{v}}(\boldsymbol{b}) - \boldsymbol{b}\|_1 \leq \Delta$, then, $F(h_{\boldsymbol{v}}(\boldsymbol{b})) = \boldsymbol{b}$. Therefore, we have:

$$
\mathbb{P}[F(h_{\boldsymbol{v}}(\boldsymbol{b})) = \boldsymbol{b}] \geq \mathbb{P}[\|h_{\boldsymbol{v}}(\boldsymbol{b}) - \boldsymbol{b}\|_1 \leq \Delta] \geq q \geq 1/2
\tag{19}
$$

By Lem. 6, for $X :\leftarrow \boldsymbol{b}$, $Y :\leftarrow h_{\boldsymbol{v}}(\boldsymbol{b})$ and $F :\leftarrow F$, we have: $I(h_{\boldsymbol{v}}(\boldsymbol{b}); \boldsymbol{b}) \geq qH(\boldsymbol{b}) - H(q)$. $\quad\square$

The following lemma is an example of three uncorrelated variables $X, Y, Z$, such that there is a dimensionality reducing linear transformation over them that preserves all of their information.

**Lemma 8.** *Let $X$ and $Y$ be two independent uniform distributions over $[-1, 1]$ and $Z = (X + Y)^2$. Then, the transformation $T(x, y, z) = (x, y)$ satisfies $I(X, Y, Z; T(X, Y, Z)) = H(X, Y, Z)$ and $\text{Cov}(X, Y) = \text{Cov}(X, Z) = \text{Cov}(Y, Z)$.*

*Proof.* Since $X$ and $Y$ are independent, their covariance is zero. By the definition of $X$ and $Y$, we have: $\mathbb{E}[X] = \mathbb{E}[Y] = \mathbb{E}[X^3] = 0$. Therefore,

$$
\begin{aligned}
\text{Cov}(Y, Z) = \text{Cov}(X, Z) &= \mathbb{E}[X(X+Y)^2] - \mathbb{E}[X]\mathbb{E}[(X+Y)^2] \\
&= \mathbb{E}[X^3] + 2\mathbb{E}[X^2]\mathbb{E}[Y] + \mathbb{E}[X]\mathbb{E}[Y^2] - \mathbb{E}[X]\mathbb{E}[(X+Y)^2] = 0
\end{aligned}
\tag{20}
$$

Finally, we consider that $T$ is a homeomorphic transformation $T : (x, y, (x+y)^2) \mapsto (x, y)$ (between the manifolds $\{(x, y, z) \mid x, y \in [-1, 1], z = (x+y)^2\}$ and $[-1, 1]^2$) and mutual information is invariant to applications of homeomorphic transformations, i.e., $I(X; Y) = I(\mu(X); \nu(Y))$ for homeomorphisms $\mu$ and $\nu$ over the sample spaces of $X$ and $Y$ (resp.). Therefore, $I(X, Y, Z; T(X, Y, Z)) = I(X, Y, Z; X, Y, Z) = H(X, Y, Z)$. $\qquad\square$

## C  PROOFS OF THE MAIN RESULTS

**Theorem 1.** *Assume that the loss function $\ell$ is symmetric and obeys the triangle inequality. Then, for any autoencoder $h = g \circ f \in \mathcal{H}$, such that $f(\boldsymbol{x}_1, \boldsymbol{x}_2) = (e_1(\boldsymbol{x}_1), e_2(\boldsymbol{x}_2)) \in \mathcal{F}$ is an encoder and $g \in \mathcal{M}$ is a decoder, the following holds,*

$$
\begin{aligned}
R_{D_{A,B}}[h, y] \leq& R_{D_{B,\hat{B}}}[h, y] + \min_{g^* \in \mathcal{M}} \left\{ R_{D_{A,B}}[g^* \circ f, y] + R_{D_{B,\hat{B}}}[g^* \circ f, y] \right\} \\
& + \text{disc}_{\mathcal{M}}(f \circ D_{A,B}, f \circ D_{B,\hat{B}})
\end{aligned}
\tag{10}
$$

*where $D_{B,\hat{B}}$ is the distribution of $(\boldsymbol{b}, \boldsymbol{b})$ where $\boldsymbol{b} \sim D_B$.*

*Proof.* Let $g^* \in \arg\min_{g \in \mathcal{M}} \left\{ R_{D_{A,B}}[g \circ f, y] + R_{D_{B,\hat{B}}}[g \circ f, y] \right\}$. Since the loss $\ell$ obeys the triangle inequality,

$$
R_{D_{A,B}}[g \circ f, y] \leq R_{D_{A,B}}[g \circ f, g^* \circ f] + R_{D_{A,B}}[g^* \circ f, y]
\tag{21}
$$

By the definition of discrepancy,

$$R_{D_{A,B}}[g \circ f, y] \leq R_{D_{B,\hat{B}}}[g \circ f, g^* \circ f] + R_{D_{A,B}}[g^* \circ f, y] + \text{disc}_{\mathcal{M}}(f \circ D_{A,B}, f \circ D_{B,\hat{B}}) \quad (22)$$

Again, by the triangle inequality,

$$\begin{aligned} R_{D_{A,B}}[g \circ f, y] \leq& R_{D_{B,\hat{B}}}[g \circ f, y] + R_{D_{A,B}}[g^* \circ f, y] + R_{D_{B,\hat{B}}}[g^* \circ f, y] \\ &+ \text{disc}_{\mathcal{M}}(f \circ D_{A,B}, f \circ D_{B,\hat{B}}) \\ =& R_{D_{B,\hat{B}}}[g \circ f, y] + \min_{g \in \mathcal{M}} \left\{ R_{D_{A,B}}[g \circ f, y] + R_{D_{B,\hat{B}}}[g \circ f, y] \right\} \\ &+ \text{disc}_{\mathcal{M}}(f \circ D_{A,B}, f \circ D_{B,\hat{B}}) \end{aligned} \quad (23)$$

$\square$

**Lemma 1.** *Let $\mathcal{M}$ be the set of neural networks of the form: $c(\boldsymbol{x}) = \phi(\boldsymbol{W}_r \ldots \phi(\boldsymbol{W}_2 \phi(\boldsymbol{W}_1 \boldsymbol{x} + \boldsymbol{q})))$, where, $\boldsymbol{W}_i \in \mathbb{R}^{d_i \times d_{i+1}}$ for $i \in \{1, \ldots, r-1\}$, $\boldsymbol{q} \in \mathbb{R}^{d_2}$ and $d_1 = E_1 + E_2$. In addition, $\phi(x_1, \ldots, x_k) = (\phi_1(x_1), \ldots, \phi_1(x_k))$, for $k \in \mathbb{N}$, $(x_1, \ldots, x_k) \in \mathbb{R}^k$ and a non-linear activation function $\phi_1 : \mathbb{R} \to \mathbb{R}$. Let $\mathcal{M}'$ be the same as $\mathcal{M}$ with $d_1 = E_1$ (instead of $d_1 = E_1 + E_2$). Let $f(\boldsymbol{x}) = (e_1(\boldsymbol{x}), e_2(\boldsymbol{x}))$ be an encoder and assume that: $e_1(\boldsymbol{b}) \perp\!\!\!\perp e_2(\boldsymbol{b})$. Then,*

$$\text{disc}_{\mathcal{M}}(f \circ D_{A,B}, f \circ D_{B,\hat{B}}) \leq \text{disc}_{\mathcal{M}'}(e_1 \circ D_A, e_1 \circ D_B) \quad (12)$$

*Proof.* By the definition of discrepancy, and since $e_1(\boldsymbol{b}) \perp\!\!\!\perp e_2(\boldsymbol{b})$, we have:

$$\begin{aligned} &\text{disc}_{\mathcal{M}}(f \circ D_{A,B}, f \circ D_{B,\hat{B}}) \\ =& \sup_{c_1,c_2 \in \mathcal{M}} \Big| \mathbb{E}_{e_1(\boldsymbol{a}),e_2(\boldsymbol{b})} \ell(c_1(e_1(\boldsymbol{a}), e_2(\boldsymbol{b})), c_2(e_1(\boldsymbol{a}), e_2(\boldsymbol{b}))) \\ &\qquad\qquad - \mathbb{E}_{e_1(\boldsymbol{b}),e_2(\boldsymbol{b})} \ell(c_1(e_1(\boldsymbol{b}), e_2(\boldsymbol{b})), c_2(e_1(\boldsymbol{b}), e_2(\boldsymbol{b}))) \Big| \\ =& \sup_{c_1,c_2 \in \mathcal{M}} \Big| \mathbb{E}_{e_2(\boldsymbol{b})} \Big\{ \mathbb{E}_{e_1(\boldsymbol{a})} \ell(c_1(e_1(\boldsymbol{a}), e_2(\boldsymbol{b})), c_2(e_1(\boldsymbol{a}), e_2(\boldsymbol{b}))) \\ &\qquad\qquad - \mathbb{E}_{e_1(\boldsymbol{b})} \ell(c_1(e_1(\boldsymbol{b}), e_2(\boldsymbol{b})), c_2(e_1(\boldsymbol{b}), e_2(\boldsymbol{b}))) \Big\} \Big| \end{aligned} \quad (24)$$

By $|\mathbb{E}[x]| \leq \mathbb{E}[|x|]$ and $\mathbb{E}_{\boldsymbol{x}}[\sup_{\boldsymbol{y} \in \mathbb{Y}} f(\boldsymbol{x}, \boldsymbol{y})] \leq \sup_{\boldsymbol{y} \in \mathbb{Y}} \mathbb{E}_{\boldsymbol{x}}[f(\boldsymbol{x}, \boldsymbol{y})]$, we have:

$$\begin{aligned} &\text{disc}_{\mathcal{M}}(f \circ D_{A,B}, f \circ D_{B,\hat{B}}) \\ \leq& \mathbb{E}_{e_2(\boldsymbol{b})} \sup_{c_1,c_2 \in \mathcal{M}} \Big| \Big\{ \mathbb{E}_{e_1(\boldsymbol{a})} \ell(c_1(e_1(\boldsymbol{a}), e_2(\boldsymbol{b})), c_2(e_1(\boldsymbol{a}), e_2(\boldsymbol{b}))) \\ &\qquad\qquad - \mathbb{E}_{e_1(\boldsymbol{b})} \ell(c_1(e_1(\boldsymbol{a}), e_2(\boldsymbol{b})), c_2(e_1(\boldsymbol{b}), e_2(\boldsymbol{b}))) \Big\} \Big| \\ \leq& \sup_{\boldsymbol{y} \in \mathbb{R}^{E_2}} \sup_{c_1,c_2 \in \mathcal{M}} \Big| \Big\{ \mathbb{E}_{e_1(\boldsymbol{a})} \ell(c_1(e_1(\boldsymbol{a}), \boldsymbol{y}), c_2(e_1(\boldsymbol{a}), \boldsymbol{y})) \\ &\qquad\qquad - \mathbb{E}_{e_1(\boldsymbol{b})} \ell(c_1(e_1(\boldsymbol{b}), \boldsymbol{y}), c_2(e_1(\boldsymbol{b}), \boldsymbol{y})) \Big\} \Big| \end{aligned} \quad (25)$$

By the definition of $\mathcal{M}$, for any $c \in \mathcal{M}$ and a fixed vector, $\boldsymbol{y} \in \mathbb{R}^{E_2}$, there is a function $u \in \mathcal{M}'$, such that for every $\boldsymbol{x} \in \mathbb{R}^{E_1}$, we have: $c(\boldsymbol{x}, \boldsymbol{y}) = u(\boldsymbol{x})$. Therefore, we can rewrite the last equation as follows:

$$\begin{aligned} &\text{disc}_{\mathcal{M}}(f \circ D_{A,B}, f \circ D_{B,\hat{B}}) \\ \leq& \sup_{u_1,u_2 \in \mathcal{M}'} \Big| \Big\{ \mathbb{E}_{e_1(\boldsymbol{a})} \ell(u_1(e_1(\boldsymbol{a})), u_2(e_1(\boldsymbol{a})) - \mathbb{E}_{e_1(\boldsymbol{b})} \ell(u_1(e_1(\boldsymbol{b})), u_2(e_1(\boldsymbol{b}))) \Big\} \Big| \\ =& \text{disc}_{\mathcal{M}'}(e_1 \circ D_A, e_1 \circ D_B) \end{aligned} \quad (26)$$

$\square$

## C.1 EMERGENCE OF DISENTANGLED REPRESENTATIONS

In this section, we employ the theory of (Achille & Soatto, 2018) in order to show the emergence of disentangled representations, when an autoencoder $h = g \circ f$ generalizes well. Our analysis shows that by learning an autoencoder such that the encoder $f$ has log-normal regularization, there is a high likelihood of learning disentangled representations. We note that until now, we treated the autoencoder $h$ as a function of two variables ($a, b$ or $b, b$). From now on, if the two variables are the same, then, we simply write $h_v(b)$. In addition, by Eq. 11, instead of writing, $R_{D_{B,\hat{B}}}[g \circ f, y]$ we can simply write $R_{D_B}[h, \mathrm{Id}] := \mathbb{E}_f \mathbb{E}_{b \sim D_B}[\ell(g \circ f(b), b)]$.

The framework of Achille & Soatto (2018) relies on a few assumptions, which are imported to our case. The algorithm is provided with $m$ i.i.d samples $\mathbb{S}_B = \{b^i\}_{i=1}^m$ from the distribution $D_B$ and trains an encoder-decoder neural network $h_v = g_u \circ f_w$, such that the parameters of the encoder, $f_v$, have log-normal perturbations. Formally, we learn a mapping $h_v$ of the form $h_v(x) = \phi(W^{2t}\phi(W^{2t-1}\ldots\phi(W^1 x)))$, where $W^i \in \mathbb{R}^{d_{i+1} \times d_i}$, for $i \in \{1, \ldots, 2t-1\}$, $d_1 = d_{2t} = M$, $w = (W^t, \ldots, W^1)$, $u = (W^{2t}, \ldots, W^{t+1})$ and $v = (W^{2t}, \ldots, W^1)$. Here, $\phi(x_1, \ldots, x_m) = (\phi_1(x_1), \ldots, \phi_1(x_k))$ is a non-linear activation function $\phi_1 : \mathbb{R} \to \mathbb{R}$ extended for all $m \in \mathbb{N}$ and $(x_1, \ldots, x_k) \in \mathbb{R}^k$. We assume that $\phi_1 : \mathbb{R} \to \mathbb{R}$ is a homeomorphism (i.e., $\phi_1$ is invertible, continuous and $\phi_1^{-1}$ is also continuous). The encoder $f_w$ is the composition of the first $t$ layers and the decoder $g_u$ is composed of the last $t$ layers of $h_v$. Following Achille & Soatto (2018), we assume that the posterior distribution $p(W_{i,j}^k | \mathbb{S}_B)$ is defined as a Gaussian dropout,

$$W_{i,j}^k | \mathbb{S}_B \sim \epsilon_{i,j}^k \cdot \hat{W}_{i,j}^k \tag{27}$$

where $\epsilon^i \in \mathbb{R}^{d_{i+1} \times d_i}$ and $\epsilon_{i,j}^k \sim \log\mathcal{N}(-\alpha/2, \alpha)$, for $k \in \{1, \ldots, t\}$. We consider that the mean and variance of $\log\mathcal{N}(-\alpha/2, \alpha)$ are 1 and $\exp(\alpha) - 1$ respectively. Here, $\hat{W}^k$ is a learned mean of the $k$'th layer of the encoder. We denote the output of the $k$'th layer of $f_w$ by $z^k$, i.e., $z^k = \phi(W^k\phi(W^{k-1}\ldots\phi(W^1 x)))$.

**Implicit Emergence of Disentangled Representations**   The following lemma is a corollary of Proposition 5.2 in (Achille & Soatto, 2018) for our model.

**Lemma 9.** *Let* $k \in \{1, \ldots, t\}$, $y^k = W^k\phi(W^{k-1}\ldots\phi(W^1 b))$ *and* $z^k = \phi(y^k)$, *for* $W^k = \epsilon^k \bigodot \hat{W}^k$, *where* $\epsilon_{i,j}^k \sim \log\mathcal{N}(\alpha_k/2, \alpha_k)$. *Further, assume that the marginals of* $\mathbb{P}[y^k]$ *and* $\mathbb{P}[y^k | z^{k-1}]$ *are Gaussians, the components of* $z^{k-1}$ *are uncorrelated and that their kurtosis is uniformly bounded. Let* $q(\alpha) := -\frac{1}{2}\log(1 - \exp(-\alpha))$. *Then,*

$$TC(z^k) \leq \dim(z^k) \cdot q(\alpha_k) - I(h_v(b); b) + \mathcal{O}\left(\frac{\dim(z^k)}{\dim(z^{k-1})}\right) \tag{28}$$

*Proof.* By Proposition 5.2 in Achille & Soatto (2018), we have:

$$\frac{TC(y^k) + I(y^k; z^{k-1})}{\dim(y^k)} \leq q(\alpha_k) + \mathcal{O}\left(\frac{1}{\dim(z^{k-1})}\right) \tag{29}$$

By Lem. 4, we have, $TC(y^k) = TC(z^k)$. In addition, $\dim(y^k) = \dim(z^k)$. Therefore,

$$TC(z^k) + I(y^k; z^{k-1}) \leq \dim(z^k) \cdot q(\alpha_k) + \mathcal{O}\left(\frac{\dim(z^k)}{\dim(z^{k-1})}\right) \tag{30}$$

Finally, by the data processing inequality, for $X_1 := b$, $X_2 := z^{k-1}$, $X_3 := y^k$ and $X_4 := h_v(b)$, we have, $I(h_v(b); b) \leq I(y^k; z^{k-1})$. The bound follows from Eq. 30 and the last observation. □

Lem. 9 provides an upper bound on the total correlation of the $k$'th layer of the autoencoder $h_v(b)$. This bound assumes that the marginal distributions of $y^k$ and $y^k | z^{k-1}$ are Gaussians. This is a reasonable assumption if $\dim(z^{k-1})$ is large by the central limit theorem. We also assume that there are no pair-wise linear correlations between the components of $z^{k-1}$. In some sense, this assumption can be viewed as minimality of $z^{k-1}$. Informally, if there is a strong linear correlation between two components of $z^{k-1}$, then, we can throw away one of them and keep most of the information. On the other hand, if the components of $z^{k-1}$ are uncorrelated, the existence of a dimensionality

reducing linear transformation $\boldsymbol{W}$ such that $I(\boldsymbol{W}\boldsymbol{z}^{k-1}; \boldsymbol{z}^{k-1}) = H(\boldsymbol{z}^{k-1})$ is still a possibility (for a concrete example, see Lem. 8). Hence, the next layer, $\boldsymbol{z}^k$, is still useful, in order to compress the representation and can still preserve all the information. We also assume that the kurtosis of the components of $\boldsymbol{z}^{k-1}$ is uniformly bounded. This is a technical hypothesis that is always satisfied, if the components of $\boldsymbol{z}^{k-1}$ are sub-Gaussian or with uniformly bounded support.

The function $q(\alpha_k)$ is monotonically increasing as $\alpha_k$ tends to 0. Therefore, this bound realizes a trade-off between the mutual information of $h_{\boldsymbol{v}}(\boldsymbol{b})$ and $\boldsymbol{b}$ and the amount of perturbations in the encoder. If the perturbations in the encoder are stronger, then, the ability of the autoencoder to reconstruct $\boldsymbol{b}$ decays. On the other hand, if the perturbations are small, then, $q(\alpha)$ increases.

We consider that the bound decreases, as $I(h_{\boldsymbol{v}}(\boldsymbol{b}); \boldsymbol{b})$ increases. It is reasonable to believe that since $h_{\boldsymbol{v}}(\boldsymbol{b})$ is an autoencoder that is being trained to reconstruct $\boldsymbol{b}$, $I(h_{\boldsymbol{v}}(\boldsymbol{b}); \boldsymbol{b})$ is maximized implicitly. The problem of training an autoencoder that maintains a reconstruction that has high mutual information with respect to the input has recently attracted a considerable attention. Several methods for maximizing this term explicitly were proposed in (Zhao et al., 2017; Phuong et al., 2018). In the following lemma, we show that if the samples of $D_B$ are well separated, then, the mutual information $I(h_{\boldsymbol{v}}(\boldsymbol{b}); \boldsymbol{b})$ is implicitly maximized as the reconstruction, $h_{\boldsymbol{v}}(\boldsymbol{b}) \approx \boldsymbol{b}$ improves.

**Lemma 10.** *Let $\boldsymbol{b} \sim D_B$ be a random vector over a discrete set $\mathbb{X}_B \subset \mathbb{R}^M$ and $h_{\boldsymbol{v}} : \mathbb{R}^M \to \mathbb{R}^M$ an autoencoder. Assume that $\forall \boldsymbol{x}_1 \neq \boldsymbol{x}_2 \in \mathbb{X}_B : \|\boldsymbol{x}_1 - \boldsymbol{x}_2\|_1 > \Delta$ and $R_{D_B}[h_{\boldsymbol{v}}(\boldsymbol{b}), \mathrm{Id}] \leq \frac{\Delta}{2}$. Then,*

$$I(h_{\boldsymbol{v}}(\boldsymbol{b}); \boldsymbol{b}) \geq \left(1 - \frac{R_{D_B}[h_{\boldsymbol{v}}(\boldsymbol{b}), \mathrm{Id}]}{\Delta}\right) H(\boldsymbol{b}) - \sqrt{R_{D_B}[h_{\boldsymbol{v}}(\boldsymbol{b}), \mathrm{Id}]} \tag{31}$$

*Proof.* First, assume by contradiction that:

$$\mathbb{P}[\|h_{\boldsymbol{v}}(\boldsymbol{b}) - \boldsymbol{b}\|_1 \leq \Delta] < \left(1 - \frac{R_{D_B}[h_{\boldsymbol{v}}(\boldsymbol{b}), \mathrm{Id}]}{\Delta}\right) \tag{32}$$

In other words, $\mathbb{P}[\|h_{\boldsymbol{v}}(\boldsymbol{b}) - \boldsymbol{b}\|_1 > \Delta] > R_{D_B}[h_{\boldsymbol{v}}(\boldsymbol{b}), \mathrm{Id}]/\Delta$. In particular, we arrive at a contradiction:

$$R_{D_B}[h_{\boldsymbol{v}}, \mathrm{Id}] = \mathbb{E}_{\boldsymbol{b}}[\|h_{\boldsymbol{v}}(\boldsymbol{b}) - \boldsymbol{b}\|_1] \geq \mathbb{P}[\|h_{\boldsymbol{v}}(\boldsymbol{b}) - \boldsymbol{b}\|_1 > \Delta] \cdot \Delta > R_{D_B}[h_{\boldsymbol{v}}, \mathrm{Id}] \tag{33}$$

By the above contradiction and the hypothesis that $\frac{R_{D_B}[h_{\boldsymbol{v}}(\boldsymbol{b}), \mathrm{Id}]}{\Delta} \leq 1/2$, we have, $q \geq \left(1 - \frac{R_{D_B}[h_{\boldsymbol{v}}(\boldsymbol{b}), \mathrm{Id}]}{\Delta}\right) \geq 1/2$. Therefore, by Lem. 7,

$$I(h_{\boldsymbol{v}}(\boldsymbol{b}); \boldsymbol{b}) \geq \left(1 - \frac{R_{D_B}[h_{\boldsymbol{v}}(\boldsymbol{b}), \mathrm{Id}]}{\Delta}\right) H(\boldsymbol{b}) - H\left(1 - \frac{R_{D_B}[h_{\boldsymbol{v}}(\boldsymbol{b}), \mathrm{Id}]}{\Delta}\right) \tag{34}$$

Additionally, by Lem. 5, we have:

$$H\left(1 - \frac{R_{D_B}[h_{\boldsymbol{v}}(\boldsymbol{b}), \mathrm{Id}]}{\Delta}\right) \leq 2\log(2)\sqrt{\left(1 - \frac{R_{D_B}[h_{\boldsymbol{v}}(\boldsymbol{b}), \mathrm{Id}]}{\Delta}\right) \cdot \frac{R_{D_B}[h_{\boldsymbol{v}}(\boldsymbol{b}), \mathrm{Id}]}{\Delta}}$$

$$\leq 2\log(2)\sqrt{\frac{R_{D_B}[h_{\boldsymbol{v}}(\boldsymbol{b}), \mathrm{Id}]}{\Delta}} \leq \sqrt{\frac{R_{D_B}[h_{\boldsymbol{v}}(\boldsymbol{b}), \mathrm{Id}]}{\Delta}} \tag{35}$$

Finally,

$$I(h_{\boldsymbol{v}}(\boldsymbol{b}); \boldsymbol{b}) \geq \left(1 - \frac{R_{D_B}[h_{\boldsymbol{v}}(\boldsymbol{b}), \mathrm{Id}]}{\Delta}\right) H(\boldsymbol{b}) - \sqrt{\frac{R_{D_B}[h_{\boldsymbol{v}}(\boldsymbol{b}), \mathrm{Id}]}{\Delta}} \tag{36}$$

$\square$

The above lemma asserts that if the samples in $D_B$ are well separated, whenever the autoencoder has a small expected reconstruction error $R_{D_B}[h_{\boldsymbol{v}}(\boldsymbol{b}), \mathrm{Id}]$, then, the mutual information $I(h_{\boldsymbol{v}}(\boldsymbol{b}); \boldsymbol{b})$ is at least a large portion of $H(\boldsymbol{b})$. Therefore, we conclude that if the autoencoder generalizes well, then, it also maximizes the mutual information $I(h_{\boldsymbol{v}}(\boldsymbol{b}); \boldsymbol{b})$. Finally, we note that we cannot directly apply Lem. 10 to bound the mutual information in Lem. 9. That is because, in Lem. 9, we assume that the components of $\boldsymbol{y}^k$ are distributed normally, which implies that the distribution of $\boldsymbol{b}$ must be continuous. On the other hand, in Lem. 10 we assume that $\boldsymbol{b}$ is distributed according to a discrete distribution. In order to reduce this friction, instead of assuming that the each component of $\boldsymbol{z}^k$ is distributed according to a normal distribution, we can assume that it is distributed according to a discrete approximation of a normal distribution.

