# OpenReview forum: "Emerging Disentanglement in Auto-Encoder Based Unsupervised Image Content Transfer"
_ICLR.cc/2019/Conference_

### Official Review · AnonReviewer3 · 2018-11-02
**Interesting Formulation/Results, Writing Can be Improved**

**Rating:** 6
**Confidence:** 2

**Review:**

This paper proposes an unsupervised style transfer method uses two-pathway encoder and a decoder for both domains. The loss function can be written using reconstruction losses and the confusion term. Experimental results are very promising comparing to state of the art methods.

The methodology presented in this paper is simple yet powerful according to the experimental results. However I do have a few concerns:

1. The writing can certainly be improved.  I had a difficult time understanding Section 2. For example the function Q is upper cased but later the f and g are all lower cased. Why domains A and B are defined using the space and the probability measure? "our framework assumes that the distribution of persons with sunglasses and that of persons without them is the same," The "distribution of persons" is not a rigorous definition and is hard to infer what does it actually mean. "f" does not appear in the loss terms although it appears under "min".

2. I like the simplicity of the objective function, but it is hard for me to understand that why the algorithm does not pick up spurious differences between A and B. For example, what if there are lighting differences and glasses/no-glasses differences between A and B? See 3rd row of figure 2 for an example.

3. Given the huge differences in performance between the proposed method and MUNIT and DRIT, some analysis/discussion on the reason of success/failure should be given.

--------------------------------------------------------

I have read authors' response.

---

> ### Author Response · Authors · 2018-11-06
> **Thank you for your review**
>
> Thank you very much for your comments and for supporting our results and framework. Below, we address the raised concerns one by one. Please let us know if you are not satisfied with our replies.
>
> Concern #1:
>
> We went to great lengths to adhere to the math style recommended provided this year by the ICLR program chairs and have employed the suggested conventions from math_commands.tex.
>
> Reviewer: The function Q is uppercased but later f and g are all lower-cased.
>
> Answer: We wanted to create a clear distinction between real networks that are learned (f,g) and an unknown underlying representation Q. We would be happy to change this.
>
> Reviewer: Why domains A and B are defined using the space and the probability measure?
>
> Answer: This is the conventional formal approach for defining a domain from which the samples are being selected i.i.d in some sample space. This is the common assumption in machine learning, e.g.,  Vapnik (2000), Bousquet and Alisseeff (2001), and it is required for the theoretical results.
>
> Reviewer: "our framework assumes that the distribution of persons with sunglasses and that of persons without them is the same," The "distribution of persons" is not a rigorous definition and is hard to infer what does it actually mean.
>
> Answer: We should have been more careful and discussed “distribution of images of persons” and not “distribution of persons.” This sentence is an example given after the rigorous definition. Specifically, we wanted to explain Eq. 2 using our running example.  The entire paragraph reads “Note that within Eq. 2, there is an assumption on the underlying distributions D_A and D_B. Using the concrete example, our framework assumes that the distribution of persons” etc.
>
> Reviewer: "f" does not appear in the loss terms although it appears under "min".
>
> Answer: f is defined 2 lines above (see Eq. 5) as f(a,b) = (e_1(a),e_2(b)) and the loss terms includes e_1 and e_2.
>
>  Concern #2:
>
> Reviewer: ...why the algorithm does not pick up spurious differences between A and B. For example, what if there are lighting differences and glasses/no-glasses differences between A and B?
>
> Answer: We divide our answer to two parts: common to many methods and specific to our method. It should be noted that while guided mapping occurs for individual images but is based on a preliminary training of unlabeled and unmatched images from the two domains.
>
> (i) Similarly to many other A to B mapping methods in the literature, the algorithm learns what differs between the domains based on the examples of the training set. Given a large enough sample size, the spurious differences are not as consistent as the target difference. In other words, using the concrete example given in the question, differences in lighting appear in both images with glasses and images without, and are therefore encoded in the common part of the representation.
>
> (ii) In our method, this effect is amplified. The representations of A and B are asymmetric and the network, by design, assigns to images in B content that is not present in A. When this content is removed, a loss (Eq. 9) ensures that we obtain images that are indistinguishable from images in A.
>
> Concern #3:
>
> Reviewer: Given the huge differences in performance between the proposed method and MUNIT and DRIT, some analysis/discussion on the reason of success/failure should be given.
>
> Answer: We explicitly mention in the paper: “The type of guiding that is obtained from the target domain in MUNIT is referred to as style, while in our case, the guidance provides content. Therefore, MUNIT, as can be seen in our experiments, cannot add specific glasses, when shifting from the no-glasses domain to the faces with eyewear domain.”
>
> In the next version, we will make sure to elaborate on this. The MUNIT and DRIT architectures lead the methods to focus on conditional style, i.e. global changes to the picture, while our method focuses on local changes (content). Therefore when given two images, MUNIT and DRIT look at the reference picture and pick up global “style” characteristics, such as background or lighting, while we are able to capture the added content.
>
> MUNIT and DRIT both use two different types of encoders that enforce a separation of the latent space representations to either style or content vectors. For example, the style encoder, unlike the content encoder, employs spatial pooling. It also results in a smaller representation than the content one. This is important, in the context of these methods, in order to ensure that the two representations encode different aspects of the image. If MUNIT/DRIT were to use the same type of encoder twice, then one encoder could capture all the information and the image-based guiding (mixing representations from two images) would become mute.
>
> In contrast, our method (i) does not separate style and content, and (ii) as mentioned above, has a representation that is geared toward capturing the additional content.

---

### Official Review · AnonReviewer2 · 2018-11-05
**Unsupervised disentanglement approach for content transfer**

**Rating:** 6
**Confidence:** 1

**Review:**

The paper proposes an unsupervised approach for mapping two sets of objects, A and B, such that set B contains all the information that is in set A and some additional information. The paper learns a latent space which encodes: (a) information which is shared in both sets, and (b) the additional content present in B. This is done by employing a two-pathway encoder and a decoder for both the sets. Experiments on problems such as adding glasses or facial hair to faces shows that the proposed method performs better than existing disentanglement approaches.

---

### Official Review · AnonReviewer4 · 2018-11-10
**interesting approach for a very specific task**

**Rating:** 6
**Confidence:** 3

**Review:**

This paper tackles the task of content transfer. For a given type of images (frontal face shots), the goal is to transfer a particular localized property (e.g. glasses or facial hair) extracted from one image to another image of the same type (difference face). This is also known as the problem of guided image-to-image translation.
The problem is formalized as the one of learning to map two different domains, one domain being composed of images with the property/attribute of interest, the other one containing images without it. The problem is said to be ‘unsupervised’, i.e. there is no pairwise correspondences between images of the two domains (with/without attributes).
The novelty of the approach lies on
-	the loss, which is composed of three terms: two reconstruction losses and a domain confusion loss
-	the overall architecture and in particular the fact that images are represented as a combination of the output of two encoders: one encodes the face and the other encodes the property (e.g. glasses).

Overall comments:
+ a theoretical part discusses generalization bounds and the emergence of disentangled representations
+ visual results are appealing showing the suitability of the method to the considered task
- the discussion of the advantages of the proposed method could be improved
- the motivation for some of the experimental results is unclear (choice of experimental protocol and baselines).
- the scope of the method seems limited

Detailed comments:

I personally like the described model. The disentanglement mechanism is intuitive to understand, and seems well suited for this particular task, as qualitative evidence suggests. I am not sure if this approach would be applicable beyond the very specific scenario considered in the paper.

The paper emphasizes that the strength of the method lies on its simplicity w.r.t. competitors, and its better results. These two aspects could be better discussed.

Simplicity:
In several places the paper claims that the proposed approach is considerably simpler. Some parts hint to criteria for the ‘complexity’ comparison, such as Table 1 or a few sentences (e.g. “this allows us to train with many less parameters and without the need to applying excessive tuning”). It would be more convincing to have a dedicated discussion of the practical advantages of the simplicity claimed by this method, discussing e.g. training/testing time, memory footprint of the models, convergence properties, stability, etc.

Comparison:
The chosen baselines, i.e. MUNIT and DRIT are experimentally shown to perform poorly on the considered task. Yet although these methods were also developed for guided image translation, they were designed for a rather different application: style transfer. I am not sure these comparisons bring much insight on the performance of the method.
Experiments are conducted for a very specific task, on a single dataset. Would the method have broader application?

Experimental protocol:
I understand that such an approach is difficult to evaluate quantitatively but I am not sure what there is to learn from experiments reported in Table 3, as there is no point of comparison on this task. This could be clarified.

Additional comments:
-	The paper relies on the assumption that the distribution of persons with sunglasses and that of persons without them is the same, except for the sunglasses. This sounds like a strong requirement for the data used to train the network; it would be interesting to discuss the practical impact of this assumption, especially on the data requirement for the method to perform well
-	I found Figure 1 quite useful. A visual representation of the architecture and its associated description help follow the technical part.
-	I got confused with some of the claims in section 4.2. More generally, I found the technical part hard to follow.
-	The user study seems small: only 10 pairs of images are considered. How were those pair chosen? Is the set representative?

---

> ### Author Response · Authors · 2018-11-11
> **Thank you for your constructive and detailed comments**
>
> Thank you very much for your supportive comments. Below, we address your comments one by one. Please let us know if this does not clarify your concerns.
>
> Specific task: the task we tackle is widely applicable. However, we tested it on images due to the availability of accessible data. Other examples in which the method can be applied include music datasets, where a musical instrument is added, computer design, where one wishes to add elements to a blueprint, the addition of certain style elements to text, and so on.
>
> The same argument used by the authors of Fader networks (NIPS, 2017) applies here: “A key advantage of our method compared to many recent models is that it generates realistic images of high resolution without needing to apply a GAN to the decoder output. As a result, it could easily be extended to other domains like speech, or text, where the backpropagation through the decoder can be really challenging because of the non-differentiable text generation process for instance.“
>
> Note that we are at least as general as the Fader networks. While they subtract content or add generic content, we allow the addition of specific ones. In order to demonstrate this, we have added to the revised version an experiment where we employ our method to remove content.
>
> More specifically, Fader networks cannot employ guidance to add glasses (or other features). Therefore, the task we compare with Fader networks on, is the one of removing glasses. In our case, we use the trained networks and follow a straightforward pipeline: we embed a picture of a person with glasses, zero the part that corresponds to glasses and decode the obtained representation. The revised version contains, in a new part of Sec. 5, qualitative results as well as a quantitative evaluation and a user study.
>
> This experiment can be seen as a specific instance of “semi-supervised source separation”, which is researched problem by itself [Smaragdis et al, "Supervised and semi-supervised separation of sounds from single-channel mixtures." In Int.  Conf. on ICA and Signal Separation, 2007].
>
> To show the applicability to music, we present here an initial result in which the two domains are Jazz music with and without percussion instruments. In the preliminary example below we show how we add the drums, as they appear in one music segment, to a musical segment without drums. This is done in the spectral domain and the quality is therefore limited.
>
> Reference source with no percussion: https://instaud.io/2Uk5
> Guide (jazz music with percussion): https://instaud.io/2UjY
> Drums from the second sample added to the first: https://instaud.io/2Uk7
>
> Reviewer:
> In several places the paper claims that the proposed approach is considerably simpler. Some parts hint to criteria for the ‘complexity’ comparison, such as Table 1 or a few sentences (e.g. “this allows us to train with many less parameters and without the need to applying excessive tuning”). It would be more convincing to have a dedicated discussion of the practical advantages of the simplicity claimed by this method, discussing e.g. training/testing time, memory footprint of the models, convergence properties, stability, etc.
>
> Answer:
> Tab. 1 compares the various methods with respect to the number of networks and the type of representation sharing used. More networks and less sharing leads to a much more complex optimization problem.
>
> Following the review, we added Tab. 2 in the revised version, which directly compares the runtime and memory footprint of each method. In addition to the results in the table, we note that we use the same hyperparameters throughout all experiments (code is publically available), which leads us to believe that the simplicity of our method results in added robustness.
>
> Reviewer:
> The chosen baselines, i.e. MUNIT and DRIT are experimentally shown to perform poorly on the considered task. Yet although these methods were also developed for guided image translation, they were designed for a rather different application: style transfer. I am not sure these comparisons bring much insight on the performance of the method.Experiments are conducted for a very specific task, on a single dataset. Would the method have broader application?
>
> Answer:
> We compare to MUNIT and DRIT since these are the closest methods in the literature. We have added above experiments comparing aspects of our method with the Fader networks.
>
> Reviewer:
> I understand that such an approach is difficult to evaluate quantitatively but I am not sure what there is to learn from experiments reported in Table 3, as there is no point of comparison on this task. This could be clarified.
>
> Answer:
> This experiment was added to further complement the user study in the previous table of the original manuscript. Since there is no method that can be used as a point of comparison (Fader networks do not use a guided image), and following the review, the table removed in the revised version.
>
> (continued below)

---

> > ### Author Response · Authors · 2018-11-11
> > **(the rest of our reply)**
> >
> > Reviewer:
> > The paper relies on the assumption that the distribution of persons with sunglasses and that of persons without them is the same, except for the sunglasses. This sounds like a strong requirement for the data used to train the network; it would be interesting to discuss the practical impact of this assumption, especially on the data requirement for the method to perform well
> >
> > Answer:
> > We did not try to enforce this requirement in any way and use all benchmarks as is.
> >
> > The assumption is required for the theoretical analysis. Without it, we would need an additional term that reflects the divergence between the distributions.
> >
> > Reviewer:
> > I got confused with some of the claims in section 4.2. More generally, I found the technical part hard to follow.
> >
> > Answer:
> > To increase the readability of Sec. 4.2, we presented each result both informally and formally. In the revised version, following the review, we have added additional clarifications. While the new text is technical, we hope that the arguments it contains are easy to follow.
> >
> > Reviewer:
> > The user study seems small: only 10 pairs of images are considered. How were those pair chosen? Is the set representative?
> >
> > Answer:
> > We considered 10 random pairs of images per each of the three transformations. The reason that we did not use more is that we wanted all users to see all the pairs (following the protocol used by CycleGAN) and did not want to have the user studies longer than they already are.
> > In the new user study (comparing to Fader networks), each user received a random subset of the test set.

---

### Author Response · Authors · 2018-11-11
**A revision and a kind request**

Dear reviewers,

We have revised the manuscript to accommodate for the comments by all reviewers. Specifically, following the comment of AnonReviewer4, a new experiment comparing to the Fader networks was added, as well as run-time statistics. We have also clarified the text where requested.

All new content is marked in red.

We would like to thank the reviewing team again and to add a request. We believe that the sentiment of the reviews is positive and that we were able to respond to all concerns with the type of results that would satisfy the reviewers. With the CVPR deadline approaching, we would appreciate an early indication by AnonReviewer3 and AnonReviewer4 on the appropriateness of our response.

Thank you,

The authors

---

### Meta-Review · Area_Chair1 · 2018-12-13
**good performance on proposed task; task not well-motivated**

**Confidence:** 3
**Recommendation:** Accept (Poster)

**Metareview:**

1. Describe the strengths of the paper.  As pointed out by the reviewers and based on your expert opinion.

The proposed method performed well on 3 visual content transfer problems.

2. Describe the weaknesses of the paper. As pointed out by the reviewers and based on your expert opinion. Be sure to indicate which weaknesses are seen as salient for the decision (i.e., potential critical flaws), as opposed to weaknesses that the authors can likely fix in a revision.

- The paper is hard to follow at times
- The problem being addressed is technically interesting but not well-motivated. That is, the question "why is this of interest to the ICLR community" was not well-answered.

3. Discuss any major points of contention. As raised by the authors or reviewers in the discussion, and how these might have influenced the decision. If the authors provide a rebuttal to a potential reviewer concern, it’s a good idea to acknowledge this and note whether it influenced the final decision or not. This makes sure that author responses are addressed adequately.

There were no major points of contention.

4. If consensus was reached, say so. Otherwise, explain what the source of reviewer disagreement was and why the decision on the paper aligns with one set of reviewers or another.

The reviewers reached a consensus that the paper should be accepted.